# Impact of prenatal phthalate exposure on newborn metabolome and infant neurodevelopment

Susan S. Hoffman[1], Ziyin Tang [2], Anne Dunlop [3], Patricia A. Brennan[4], Thompson Huynh[2], Stephanie M. Eick[2], Dana B. Barr[2], Blake Rushing[5,6], Susan L. McRitchie[6], Susan Sumner[5,6], Kaitlin R. Taibl[2], Youran Tan [2], Parinya Panuwet[2], Grace E. Lee[2], Jasmin Eatman[2], Elizabeth J. Corwin[7], P. Barry Ryan[2], Dean P. Jones [8] & Donghai Liang [1,2] ✉

We evaluated associations among exposure to prenatal phthalate metabolites, perturbations of the newborn metabolome, and infant neurobehavioral functioning in mother-newborn pairs enrolled in the Atlanta African American Maternal-Child Cohort during 2016–2018. We quantified eight phthalate metabolites in prenatal urine samples collected between 8- and 14-weeks' (visit 1; $n = 216$) and 24- and 30-weeks' gestation (visit 2; $n = 145$) and metabolite features in newborn dried-blood spot samples collected at delivery. Associations between phthalate metabolite concentrations and metabolic feature intensities at both visits were examined using adjusted generalized linear models (MWAS). Then, an exploratory meet-in-the-middle (MITM) analysis was conducted in a subset with NICU Neonatal Neurobehavioral Scale (NNNS) scores (visit 1 $n = 81$; visit 2 $n = 71$). In both the MWAS and MITM, many of the confirmed metabolites are involved in tyrosine and tryptophan metabolism, including tryptophan, tyrosine, thyroxine, and serine. This analysis elucidates how prenatal phthalate exposure disrupts the newborn metabolome and infant neurobehavioral outcomes.

Phthalates are a group of chemicals invented in 1907 and used to manufacture an extensive range of products, including vinyl flooring, soaps, shampoos, and oil. These additives make products and plastics more durable by improving flexibility or retaining color, scent, or shine, and have become integral to modern human life[1]. In 2020, the increasing demand for plastic created a $10 billion market for phthalic anhydride, the principal precursor of phthalates, despite growing concern over the health and safety of the product. As these chemicals are ubiquitous, people are routinely exposed as they interact with their environment[2–4]. Exposures occur through eating and drinking products that have come in contact with the chemical and through the use of personal care products, cosmetics, perfumes, and clothing that contain the additive. Exposure also occurs through inhalation of dust or particles contaminated with phthalates[1]. After entering the body, phthalates are metabolized rapidly and excreted in urine, and therefore, recent exposure can be assessed through measurement of primary and secondary urinary metabolite levels (Table S1)[2]. Most individuals in the United States (US) population, if not all, have detectable levels of phthalates[3]. There are differences in exposure level depending on a person's demographic characteristics, with women

[1]Department of Epidemiology, Emory University, Atlanta, GA, USA. [2]Gangarosa Department of Environmental Health, Emory University, Atlanta, GA, USA. [3]Department of Gynecology and Obstetrics, School of Medicine, Emory University, Atlanta, GA, USA. [4]Department of Psychology, Emory University, Atlanta, GA, USA. [5]Department of Nutrition, University of North Carolina at Chapel Hill, Chapel Hill, NC, USA. [6]Nutrition Research Institute, University of North Carolina at Chapel Hill, Chapel Hill, NC, USA. [7]School of Nursing, Columbia University, New York, NY, USA. [8]School of Medicine, Emory University, Atlanta, GA, USA. ✉e-mail: donghai.liang@emory.edu

more likely to have higher urinary concentrations than men[3] and Black individuals having higher urinary concentrations than white individuals[3,4]. Further, Black women generally had higher cumulative exposures to phthalates compared to white women[4,5]. This is likely due, at least in part, to differences in the use of personal care products (e.g., hair products), stemming from environmental racism, including the role of discriminatory marketing practices[6,7].

Phthalates are considered endocrine-disrupting chemicals (EDCs) and can interrupt or interact with human hormones[8]. So, although humans are exposed to low levels when interacting with the environment, these chemical compounds, like hormones, can still have an impact on health (non-monotonic toxicity)[8,9]. The widespread use of the chemicals across a myriad of daily products and the high potential for exposure have raised concerns about the impact on human health. Research has demonstrated that phthalate exposures are associated with different health outcomes[10], but most significantly impact reproductive and child health[11]. It is especially concerning that phthalates can cross the placental barrier and affect normal embryonic and fetal development[12]. Previous research has shown a link between prenatal urinary phthalate levels and impacts on neurobehavioral development in low-risk infants[13–15]. Specifically, prenatal exposure to phthalates and its interaction with the maternal endocrine system has been associated with Apgar scores 5 min after delivery[1–15]. Further, Yolton et al. found that prenatal di(2-ethylhexyl) phthalate (DEHP) metabolite levels measured at 26 weeks gestation were associated with nonoptimal reflexes, measured using NICU Network Neurobehavioral Scale (NNNS) scores, in 5-week-old male infants in a cohort of 350 mother/infant pairs[13]. Similarly, Kim, et al. found that DEHP metabolite levels measured in the third trimester were inversely associated with mental and psychomotor indices (Bayley Scales of Infants Development 2nd edition) measured at 6 months old in a cohort of 460 expectant mothers[14]. Understanding how intermediate biomarkers might underpin these associations could inform causal links or serve as metabolic markers for early intervention.

Despite these potential impacts on maternal and child health, only a handful of studies have focused on phthalate-associated effects on maternal metabolism while pregnant. One study examined the effects of phthalates on the maternal metabolome, analyzing serum samples obtained during the third trimester, as well as the impact of phthalates on the placental metabolome[16]. This research did not see an association between phthalates and maternal serum samples but did see perturbations in a mixtures analysis between phthalates and the placental metabolome, with reduced concentrations of 2-hydroxybutyrate, carnitine, O-acetylcarnitine, glucitol, and N-acetylneuraminate[16]. Another study looked at how phthalate levels were associated with the metabolome of pregnant African Americans[17]. Results indicated perturbations in biological pathways and individual metabolites involved in inflammation, oxidative stress, and endocrine disruption. Although these studies have contributed significantly to our understanding of phthalate-related changes in maternal and placental metabolomes, there is no prior study describing how prenatal urinary phthalate levels relate to the newborn metabolome. Evaluating the biological mechanisms linking prenatal exposure to phthalates with alterations in the newborn metabolome is crucial for elucidating early-life metabolic perturbations associated with these compounds and the corresponding health effects. This is particularly important given the well-documented health impacts of phthalates, especially in underserved and vulnerable populations.

In a well-established prospective cohort of pregnant African American people and their infants, we first sought to investigate the relationship between prenatal urinary phthalate levels and perturbations in the newborn metabolome. Based on these initial metabolomics results and previous epidemiological research[13–15], we also hypothesized that urinary phthalate levels may interfere with infant

neurodevelopment (evaluated by NNNS scores) through perturbations in the newborn metabolome. To test this hypothesis, we conduct an exploratory metabolomics analysis to further examine the potential mechanisms behind prenatal exposure to phthalates and neurobehavioral outcomes by identifying potential intermediate biomarkers between phthalate exposure and NNNS scores within a subset of the same population using a meet-in-the-middle (MITM) approach.

## Results
### Study population and exposure levels
In the main metabolomic-wide association study (MWAS), we evaluated a total of 216 participants with urinary phthalate data from prenatal visit 1(8-14 weeks gestation) and 145 participants with urinary phthalate data from prenatal visit 2 (24–30 weeks gestation). In the subset used for the MITM analysis (which required newborn metabolome and NNNS data), there were 81 participants with phthalate exposure data from prenatal visit 1 and 71 with phthalate exposure data from prenatal visit 2. Characteristics of participants with data from across the different visits and between the full sample MWAS and the subset were similar (Table 1). Characteristics from both subsets did not

**Table 1 | Characteristics of the sample population**

| Characteristic | Visit 1 (8–14 weeks) | | Visit 2 (24–30 weeks) | |
|---|---|---|---|---|
| | Main sample = 216 | Subset = 81 | Main sample = 145 | Subset = 71 |
| | Median (IQR) | | Median (IQR) | |
| Age at enrollment | 24 (21.0, 29.0) | 24.0 (21.0, 29.0) | 24 (21.0, 29.0) | 24 (21.0, 29.0) |
| Pre-pregnancy BMI (kg/m²) | 28 (23, 34) | 27 (24, 34) | 26 (22, 33) | 26 (23, 34) |
| Gestational age | 11.29 (9.29, 13.00) | 11.57 (9.29, 13.29) | 25.43 (24.57, 27.57) | 25.43 (24.36, 27.36) |
| Creatinine (mg/dL) | 178 (122, 228) | 180 (128, 229) | 139 (100, 202) | 143 (100, 205) |
| | n (%) | | n (%) | |
| Education | | | | |
| <High school | 34 (16) | 11 (14) | 20 (14) | 8 (11) |
| High school | 91 (42) | 35 (43) | 58 (40) | 31 (44) |
| Some college or more | 91 (42) | 35 (43) | 67 (46) | 32 (45) |
| Parity | | | | |
| No prior birth | 92 (43) | 33 (41) | 61 (42) | 30 (42) |
| Prior birth | 124 (57) | 48 (59) | 84 (58) | 41 (58) |
| Alcohol use | | | | |
| No | 194 (90) | 72 (89) | 130 (90) | 63 (89) |
| Yes | 22 (10) | 9 (11) | 15 (10) | 8 (11) |
| Tobacco use | | | | |
| No | 177 (82) | 69 (85) | 123 (85) | 62 (89) |
| Yes | 39 (18) | 12 (15) | 22 (15) | 9 (13) |
| Marijuana use | | | | |
| No | 128 (59) | 51 (63) | 89 (61) | 45 (63) |
| Yes | 88 (41) | 30 (37) | 56 (39) | 26 (37) |
| Sex of baby | | | | |
| Male | 102 (47) | 43 (53) | 68 (47) | 35 (49) |
| Female | 114 (53) | 38 (47) | 77 (53) | 36 (51) |

The main sample was used for the metabolomic-wide association study (MWAS) and analyzed independently at two different time points (visit 1 n = 216; visit 2 n = 145). The subset was used for the meet-in-the-middle analysis, analyzed separately at the two study visits (visit 1 subset n = 81; visit 2 subset n = 71). BMI body mass index (calculated as weight in kilograms divided by height in meters squared).
Atlanta African American Maternal-Child cohort, 2016–2018.

**Table 2 | Exposure information for visit 1 (n = 216) and visit 2 (n = 145) for the participants in the metabolomic-wide association study**

| Metabolite (ng/ml) | Visit 1 (8–14 weeks) n = 216 | | Visit 2 (24–30 weeks) n = 145 | | Fourth national report on human exposure to environmental chemicals (2015–2016) | |
|---|---|---|---|---|---|---|
| | Median (IQR) | GM | Median (IQR) | GM | Age ≥20 GM | Non-Hispanic black GM |
| Monoethyl phthalate (MEP) | 98 (44, 233) | 103 | 81 (34, 180) | 83.1 | 35.9 | 54.9 |
| Mono-n-butyl phthalate (MBP) | 11 (5, 20) | 9.8 | 10 (5, 26) | 9.2 | 9.36 | 10.9 |
| Monoisobutyl phthalate (MiBP) | 10 (4, 19) | 8.8 | 8 (3, 16) | 7.5 | 8.03 | 9.29 |
| Monobenzyl phthalate (MBzP) | 6 (2, 13) | 5.7 | 5 (2, 11) | 4.9 | 3.87 | 5.57 |
| Mono-2-ethylhexyl phthalate (MEHP) | 1.7 (0.6, 4.6) | 1.7 | 2.4 (0.7, 5.0) | 1.8 | # | 1.27 |
| Mono-2-ethyl-5-oxohexyl phthalate (MEOHP)* | 3.1 (1.7, 6.5) | 3.3 | 3.7 (1.5, 7.1) | 3.1 | 3.36 | 3.66 |
| Mono-2-ethyl-5-hydroxyhexyl phthalate (MEHHP)* | 5 (3, 11) | 5.3 | 5 (2, 9) | 4.2 | 5.40 | 5.96 |
| Mono-2-ethly-5-carboxypentyl phthalate (MECPP)* | 7 (4, 13) | 8.1 | 6 (4, 11) | 7.1 | 8.28 | 8.51 |
| Sum of Di-2-ethylhexyl phthalate metabolites (ΣDEHP) | 0.06 (0.03, 0.11) | 0.07 | 0.06 (0.03, 0.11) | 0.06 | - | - |

Note: All phthalate metabolites were found in 100% of samples.
*Secondary metabolite.
*GM* geometric mean, # not calculated: proportion of results below the limit of detection was too high to provide a valid result, *IQR* interquartile range.
Phthalate exposure metabolite levels (ng/ml) were compared to the Centers for Disease Control and Prevention's Fourth National Report on Human Exposure to Environmental Chemicals (2015–2016)[18]. Atlanta African American Maternal-Child cohort, 2016–2018.

differ from the larger Atlanta African American Maternal-Child Cohort (Table S2). Participants were in their mid-20s, and most had a high school education or higher. For most of the participants (57%), the pregnancy captured in this study was not their first. Participants did report substance use during pregnancy, with ~10% reporting alcohol use, ~15% reporting tobacco use, and ~40% reporting marijuana use.

The phthalate metabolite levels were summarized for each visit in the full sample (Table 2) and in the subset (Table S3). All phthalate metabolites were found in 100% of samples. In the full sample, urinary phthalate levels were compared to the geometric mean levels reported by the National Health and Nutrition Examination Survey (NHANES) during the same study period (2015-2016) in both the general US population aged 20 years or older and in non-Hispanic Blacks[18]. For MEP, the geometric mean levels at both clinical visits were higher than the levels of people aged 20+ and non-Hispanic Blacks. The geometric mean level of MEHP at both clinical visits was higher than the levels reported among non-Hispanic Blacks in the NHANES data. The other phthalate levels were comparable to the NHANES levels. Correlations of phthalate metabolites within and across visit are summarized for the full sample (Figure S3). Within visits, only phthalate metabolites that are related to DEHP (MEHP, MEOHP, MEHHP, and MECPP) were highly correlated. Across visits, phthalate metabolites were not highly correlated.

### High-resolution metabolomic profiling
After data quality filtering, 13,680 peaks were filtered out, and 7191 total metabolic features remained. In the MWAS with visit 1 at $p < 0.05$, there were 224, 687, 282, 349, 501, 289, 292, 394, and 365 features associated with MEP, MBP, MiBP, MBzP, MEHP, MEOHP, MEHHP, MECPP, and ΣDEHP, respectively (Table S4). In the MWAS with visit 2 at $p < 0.05$, there were 260, 354, 634, 304, 349, 426, 304, 488, and 393 features associated with MEP, MBP, MiBP, MBzP, MEHP, MEOHP, MEHHP, MECPP, and ΣDEHP, respectively (Table S4). These significant features were used for the pathway analysis. There were fewer features associated with urinary phthalate levels after correcting for multiple testing ($q < 0.2$; Table S4). These features were used in chemical annotation.

The metabolites associated with each phthalate exposure and either attention or arousal are summarized in Table S5. Generally, <50 metabolomic features met the threshold ($p < 0.05$), and visit 2 had more significant overlapping features than visit 1. These features were used in the MITM analysis chemical annotation.

### Pathway analysis
Using *mummichog*[19], we investigated the enrichment of endogenous metabolic pathways with features significantly associated with each phthalate metabolite ($p < 0.05$). There were 28 perturbed metabolic pathways associated with phthalate measured in visit 1 and 12 in visit 2. Eleven pathways were identified in both visits (Fig. 1). Many of the pathways identified were involved in oxidative stress, acute inflammatory response, and low-grade inflammation, such as vitamin metabolism (vitamin K, H, B6, B5, E, and B3), tyrosine metabolism, tryptophan metabolism, the urea cycle, and pyruvate metabolism (Figs. 1 and S4).

### Chemical identification
Newborn metabolomic signals were identified by comparing them to North Carolina's Human Health Exposure Analysis Resource (HHEAR) Hub's in-house experimental standards library (IESL). After correcting for multiple testing ($q < 0.2$), 24 confirmed metabolites of the newborn metabolome associated with prenatal visit 1 phthalate metabolites and six confirmed newborn metabolites associated with prenatal visit 2 phthalate metabolites. In visit 1 analyses, the majority (21 metabolites) were associated with MBP, and the remaining were associated with MEHP. In visit 2 analyses, one newborn metabolite was associated with MBP, three with MECPP, and two with both MECPP and MEOHP (Table 3). Annotated endogenous metabolites seemed to be generally involved in neurological health through perturbations in metabolites and pathways involved in tryptophan metabolism, linoleic acid metabolism, and tyrosine metabolism (Table 3; Fig. 2).

In the MITM analysis, no significant signals were associated with both exposure and outcome after correcting for multiple testing ($q < 0.2$). At a $p < 0.05$ threshold, there were 12 confirmed newborn metabolites for visit 1 and 21 identified newborn metabolites for a visit 2 that were associated with at least one phthalate metabolite and one of the NNNS outcomes. Annotated endogenous metabolites, including tyrosine, 5-hydrotyptophan, serotonin, serine, and phenylalanine, were involved in neurological health and inflammation through perturbations in tryptophan metabolism, glycine, serine, and threonine metabolism, and fatty acids (Table 4, Fig. 3).

## Discussion
Using cutting-edge, targeted exposure assessment and untargeted metabolomics methodologies, this study identified and characterized associations among in utero exposure to phthalates, perturbations of

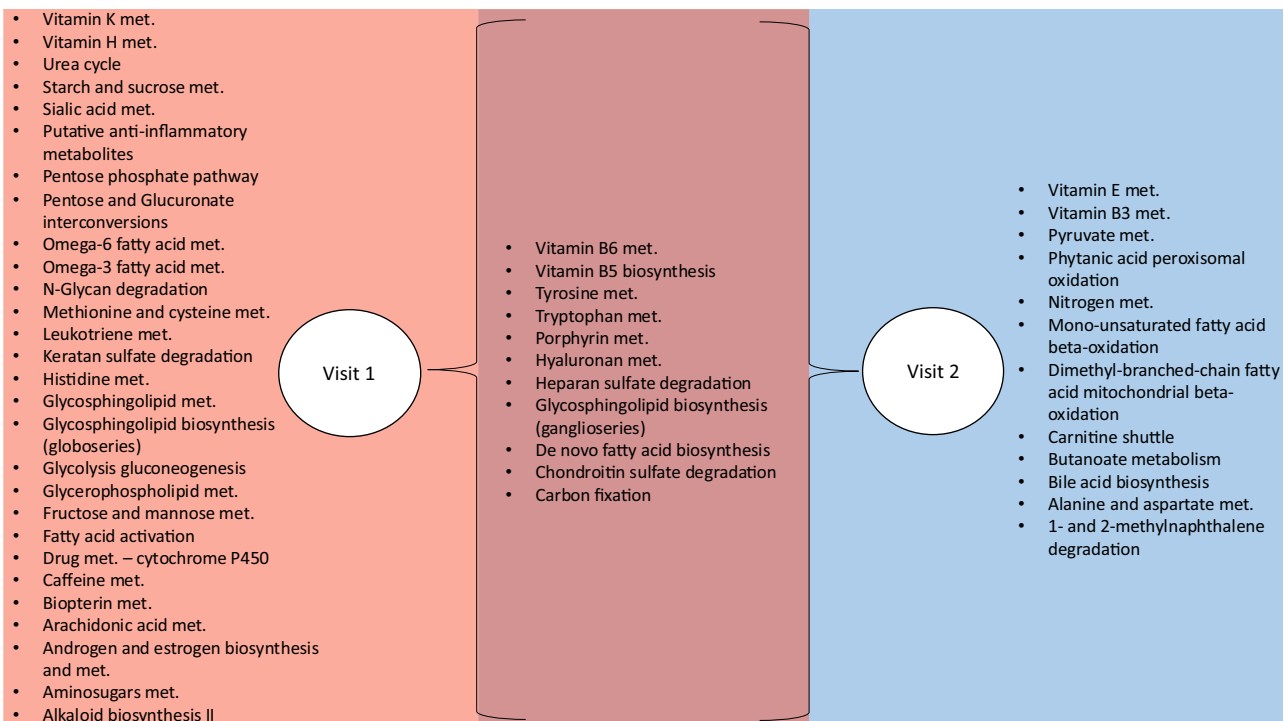

**Fig. 1 | Biological pathways associated (*p* < 0.05) with at least one phthalate metabolite showing the pathways unique to visit 1 (28) and visit 2 (12) and the pathways shared between the two visits (11).** Visit 1 *n* = 216 and visit 2 *n* = 145. Pathway analyses were conducted using the bioinformatics tool *mummichog*. Source data are provided as a Source Data file. Atlanta African American Maternal-Child cohort, 2016–2018. met metabolism.

the newborn metabolome, and measures of infant neurodevelopment. To our knowledge, this is the first study to use newborn DBS metabolomics to understand how prenatal exposure to phthalates perturb the newborn metabolome and how these perturbations could provide a molecular mechanism for impacts on NNNS scores.

Compared to NHANES, the geometric mean levels of MEP and MEHP at both clinical visits were higher in our study population than the levels of US adults (20 and older) as well as US non-Hispanic Blacks. Of note, MEP is the more abundant phthalate metabolite generally found in populations[18]. Taken together, these results support the existence of environmental disparities in phthalates exposures and suggest that we need further regulation on phthalates targeted at pregnant people generally and especially pregnant African American people.

Many of the pathways identified in the full study sample were involved in oxidative stress and acute and systemic inflammatory response, including vitamin K, H, B6, B5, E, and B3 metabolism, tyrosine metabolism, and tryptophan metabolism (Figs. 1 and S4). Further, there were more unique pathways associated with prenatal visit 1 than prenatal visit 2, suggesting that the timing of the exposure in gestation might have differential biological impacts (Fig. 1). For example, pathways associated with visit 1 only include leukotriene and arachidonic metabolism, which are highly inflammatory[20].

These findings, along with the observed variances in metabolite perturbations across visits 1 and 2 and the differential associations of various phthalate metabolites with distinct gestational stages, suggest potential implications for fetal development. In early pregnancy (visit 1), major development of the central nervous system, eyes, teeth, genitalia, and refinement of the limbs and heart are taking place. Whereas later pregnancy (visit 2) primarily entails refinement processes within these systems, particularly focusing on central nervous system maturation and refinement within the eyes and genitalia[21]. The absence of significant impacts on later gestational periods does not preclude potential effects during the critical phases of early

embryogenesis. These findings underscore the necessity for targeted investigations spanning the entirety of gestation and advocate for nuanced consideration of pregnancy timing when evaluating environmental exposures.

Based on the annotated metabolites in the full sample, we hypothesized three potential underlying mechanisms (Table 3, Fig. 2). In line with the pathway results, many of the annotated metabolites centered around tyrosine and tryptophan metabolism and are involved in abnormal neurodevelopment and brain maturation. Among the annotated metabolites, tyrosine stands out as a crucial neurotransmitter and, in this study, was associated with downregulation after exposure to phthalates[22]. Serine plays a key role as a neurotransmitter precursor, and tryptophan is essential to brain development[23,24]. Both were associated with an upregulation in response to MBP exposure, potentially as a protective response. Additionally, the presence of thyroxine, a thyroid hormone, is noteworthy, as low levels of this hormone have been associated with increased susceptibility to illness and neurodevelopmental issues in newborns[25]. This was downregulated in response to MBP exposure. Finally, the metabolite dihomo-gamma-linoleate within linoleic acid metabolism was associated with a downregulation by MBP exposure. This polyunsaturated fatty acid is a component of neural membranes[26]. Its significance for newborn health is emphasized by its abundance in breastmilk[27]. The interplay of metabolites within these pathways emphasizes the potential impact of phthalates, specifically MBP, on abnormal neurodevelopment, underlining the need for closer examination of this exposure and its intricate biochemical influences on brain maturation during this critical developmental period.

We observed similar and consistent findings in the same cohort when examining disruptions to the maternal metabolome in response to urinary phthalate levels previously[17]. Specifically, we observed biological perturbations in the maternal metabolome centering around tyrosine metabolism and also identified perturbations to tyrosine and thyroxine[17]. These robust and consistent observations underscore the

**Table 3 | Chemical identification of the metabolic features associated with phthalate exposure in the metabolic-wide association study at visit 1 (*n* = 216) and visit 2 (*n* = 145) after correction for multiple testing (*q* < 0.2) using generalized linear models**

| m/z | rt | Ontology level* | Metabolite | Exposure | Association with exposure (*β*) | Pathways | Biological function |
|---|---|---|---|---|---|---|---|
| | | | | | Visit 1 (8–14 weeks) *n* = 216 | | |
| 181.0733 | 1.62 | OL2a | Tyrosine | MBP | −0.0046 | Tyrosine metabolism (endogenous) | Thyroid function[72]; neurotransmitter production[22] |
| 211.0844 | 2.86 | OL2a | 2-amino-3-(4-hydroxy-3-methoxyphenyl) propanoic acid | MBP | 0.0030 | Pharmaceutical (foodome) | Not enough information |
| 178.0496 | 3.97 | OL2a | 3-Hydroxyhippuric acid | MBP | 0.0012 | Microbial metabolite (foodome) | Not enough information |
| 174.0549 | 6.39 | OL2b | Quinaldic acid | MBP | 0.0031 | Tryptophan metabolism (endogenous) | Not enough information |
| 243.1491 | 7.45 | OL1 | Tiglylcarnitine | MBP | 0.0049 | Acyl carnitine metabolism (endogenous) | Not enough information |
| 234.0778 | 6 | OL2a | (R)-N-Acetyl-S-(1-(hydroxymethyl)-2-propen-1-yl)-L-cysteine | MBP | −0.0032 | Acrylamide metabolite (environmental) | Acrylamide is classified as a possible human carcinogen; found in cigarettes or processed foods[32] |
| 105.0335 | 6.69 | OL1 | 4-hydroxybenzaldehyde | MBP | 0.0033 | Biosynthesis of phenyl-propanoids (endogenous) | Promotes antiparasitic host response[73] |
| 88.0393 | 2.71 | OL2b | Serine | MBP | 0.0021 | Glycine, serine and threonine metabolism (endogenous) | Neurotransmitter precursor[23] |
| 289.2523 | 16.54 | OL2a | Dihomo-gamma-linolenic acid | MBP | −0.0012 | Linoleic acid metabolism (endogenous) | Neural membrane phospholipid[26] |
| 330.2554 | 13.25 | OL2b | Docosapentaenoic acid | MBP | −0.0009 | Fatty acid (endogenous) | Anti-inflammatory; found in high concentrations in breastmilk[74] |
| 243.1127 | 7.6 | OL2a | Oxadixyl | MBP | −0.0022 | Fungicide used in food crops (environmental) | Not enough information |
| 215.1178 | 11.1 | OL2a | Melatonin | MBP | 0.0059 | Circadian entrainment (endogenous) | Anti-oxidative effects in newborns[75] |
| 190.1073 | 8.53 | OL2a | 2-Octenedioic acid | MBP | −0.0018 | Fatty acid (endogenous) | Not enough information |
| 360.2741 | 11.69 | OL1 | 3-hydroxydodecanoyl carnitine | MBP | −0.0019 | Acyl carnitine metabolism (endogenous) | Related to fatty acid oxidation and energy metabolism[76] |
| 193.0737 | 4.28 | OL2b | 3-Methylhippuric acid | MBP | 0.0014 | Xylene metabolite (environmental) | Xylenes are often used as solvents in household products[77] |
| 777.6937 | 11.82 | OL1 | Thyroxine | MBP | −0.0011 | Tyrosine metabolism (endogenous) | Thyroid hormone; low level associated with increased illness and neurodevelopmental issues in newborns[25] |
| 259.1073 | 8.58 | OL2a | Linagliptin | MBP | −0.0014 | Dipeptidyl peptidase-4 (DPP-4) inhibitor (drug) | Lowers blood sugar in patients by signaling the liver to stop producing glucose[78] |
| 193.0972 | 1.55 | OL1 | Trans-3'-Hydroxycotinine | MBP | 0.0048 | Nicotine metabolite (endogenous) | Indicator of exposure to cigarette smoke; inversely associated with birth weight[31] |
| 157.0608 | 3.3 | OL2b | Formiminoglutamic acid | MBP | 0.0023 | Folate bioavailability (foodome) | Metabolite accumulates with inadequate availability of folate; used as a biomarker of folate deficiency[79] |
| 197.0185 | 4.86 | OL2a | hydroxyphenylacetic acid | MBP | 0.0022 | Tyrosine metabolism (endogenous) | Not enough information |
| 308.0908 | 8.35 | OL2b | Glutathione reduced | MBP | 0.0017 | Glycine, serine and threonine metabolism (endogenous) | Prevents oxidative stress[80] |
| 156.1019 | 5.87 | OL2b | Acetylleucine | MEHP | 0.0061 | Valine, leucine and isoleucine metabolism (endogenous) | Decreased neural inflammation in mouse models[81] |
| 199.0962 | 7.23 | OL2a | Phenyllactic acid | MEHP | −0.0057 | Phenylalanine metabolism (endogenous) | Not enough information |
| 273.1844 | 10.95 | OL2a | Estradiol-17alpha | MEHP | 0.0080 | Steroid hormone biosynthesis (endogenous) | Reduced chronic inflammation in mouse models[82] |

**Table 3 (continued) | Chemical identification of the metabolic features associated with phthalate exposure in the metabolic-wide association study at visit 1 (n = 216) and visit 2 (n = 145) after correction for multiple testing (q < 0.2) using generalized linear models**

| m/z | rt | Ontology level* | Metabolite | Exposure | Association with exposure (β) | Pathways | Biological function |
|---|---|---|---|---|---|---|---|
| | | | | | Visit 2 (24–30 weeks) n = 145 | | |
| 292.2348 | 16.08 | OL2a | 11-Octadecen-9-ynoic acid | MBP | −0.0113 | Fatty acid (endogenous) | Not enough information |
| 237.1232 | 4.38 | OL2a | Tryptophan | MECPP | 0.0044 | Tryptophan metabolism (endogenous) | Essential to brain development and the establishment of sleep-wake cycles in newborns[24] |
| 218.1386 | 5.74 | OL2a | 15-Valerolactone | MECPP | −0.0039 | Not enough information (endogenous) | Not enough information |
| 199.1076 | 8.63 | OL2a | Tyrosine | MECPP | −0.0063 | Tyrosine metabolism (endogenous) | Thyroid function[72]; Neurotransmitter production[22] |
| 266.136 | 7.83 | OL2a | 5-methyl-1,3-benzenediol | MECPP, MEOHP | 0.0058, 0.0100 | Not enough information (foodome) | Not enough information |
| 156.1019 | 5.87 | OL2b | Acetylleucine | MECPP, MEOHP | 0.0029, 0.0055 | Valine, leucine and isoleucine metabolism (endogenous) | Decreased neural inflammation in mouse models[81] |

*MEP monoethyl phthalate, MBP mono-n-butyl phthalate, MiBP monoisobutyl phthalate, MBzP monobenzyl phthalate, MEHP mono-2-ethylhexyl phthalate, MEOHP mono-2-ethyl-5-oxohexyl phthalate, MEHHP mono-2-ethyl-5-hydroxyhexyl phthalate, MECPP mono-2-ethyl-5-carboxypentyl phthalate, ΣDEHP Sum of Di-2-ethylhexyl phthalate metabolites.*

*Ontology level indicates the strength of the evidence supporting each metabolite's identification or annotation, considering retention time, exact mass (MS), MS/MS fragmentation pattern, and isotopic ion pattern. Metabolites that closely matched the IESL by RT (within ±0.5 min), MS (within <5 ppm), and displayed a high degree of similarity in MS/MS fragmentation patterns (similarity score >30) received an OL1 label. For metabolites matching by RT and MS, they were labeled OL2a. An OL2b label was assigned to signals that matched the IESL by MS and MS/MS but fell outside the RT window (±0.5 min).

The chemical identity was conducted by matching peaks by accurate mass and retention time to authentic reference standards in an in-house library run under identical conditions using tandem mass spectrometry. Source data and exact two-sided p-values are provided as a Source Data file. Atlanta African American Maternal–Child cohort, 2016–2018.

potential interconnectedness of maternal and fetal metabolic responses to environmental exposures, emphasizing the need for a comprehensive understanding of these perturbations for both maternal and infant health.

As many of the endogenous annotated metabolites were involved with neurological function, an exploratory MITM analysis was conducted in a smaller subset of the participants with infant NNNS scores available. Many of the annotated endogenous metabolites from the MITM analysis were also part of tryptophan and tyrosine metabolism (Table 4, Fig. 3). Within tyrosine metabolism, phthalate metabolites were associated with a downregulation of tyrosine and a subsequent negative association with attention scores. This connection is especially pertinent considering that attention and arousal, key symptoms associated with clinical tyrosine deficiency and deficits in amino acids within the tyrosine metabolic pathway, play a crucial role in neurological function. Phthalate metabolites were associated with a downregulation of 5-hydrotryptophan, a precursor of the neurotransmitter serotonin, as well as serotonin itself[28,29], both of which are also part of tryptophan metabolism. Both 5-hydroxytryptophan and serotonin were associated with a decrease in arousal scores. Serine and phenylalanine, both neurotransmitter precursors[23,30], were associated with a downregulation in response to urinary phthalate levels and then subsequently related to a negative association with attention scores. The observed associations between phthalate metabolites and downregulation of key neurotransmitter precursors, coupled with negative impacts on attention and arousal scores, suggest that phthalates may have long-term developmental consequences for these infants. This study highlights the importance of further targeted research into tryptophan and tyrosine metabolism in both infants and children to understand their role in development.

We also identified and confirmed exogenous environmental metabolites from both the MWAS and the MITM analysis (Table 4). This underscores the capability of metabolomics to identify novel environmental exposures simultaneously, highlighting its utility in uncovering novel factors that may impact maternal and newborn health. For example, a fungicide (oxadixyl) was identified in the MWAS, and pesticides (propham and flonicamid) were identified in the MITM analysis. The interaction of these exposures with phthalates could be considered in future analyses. Further, biomarkers commonly associated with cigarette smoking were identified in both the MWAS (trans-3'-hydroxycotinine and (R)-N-Acetyl-S-(1-(hydroxymethyl)-2-propen-1-yl)-L-cysteine) and the MITM (cotinine, N-Acetyl-S-(3-hydroxypropyl-1-methyl)-L-cysteine, and (R)-N-Acetyl-S-(1-(hydroxymethyl)-2-propen-1-yl)-L-cysteine) analysis[31–34]. In a separate MITM study within the same cohort that investigated the effects of maternal cotinine on the maternal metabolome, serine emerged as a perturbed metabolite[35]. This finding aligns with the present study, as serine was identified in both the MWAS and MITM analysis. It is important to consider the interpretation of these findings within the context of exposure to complex mixtures. When conducting our statistical modeling, we considered each phthalate independently as a surrogate of exposure. Thus, an observed metabolic perturbation associated with a particular phthalate may not necessarily indicate a causal association between the metabolic feature and that specific modeled indicator. Instead, such change may be more likely associated with multiple, correlated pollutants within a complex exposure mixture. This consistency in perturbed metabolites across different studies with varied single exposures suggests that future research should consider the cumulative impact of multiple exposures, integrating the analysis of a mixture approach[36–38]. By doing so, we can enhance our comprehension of how multiple components of the exposome collectively influence health outcomes.

We also identified metabolites in the MWAS, including 3-hydroxyhippuric acid and hydroxyphenylacetic acid, which are related to the microbiome[39,40]. Metabolomics has the potential to

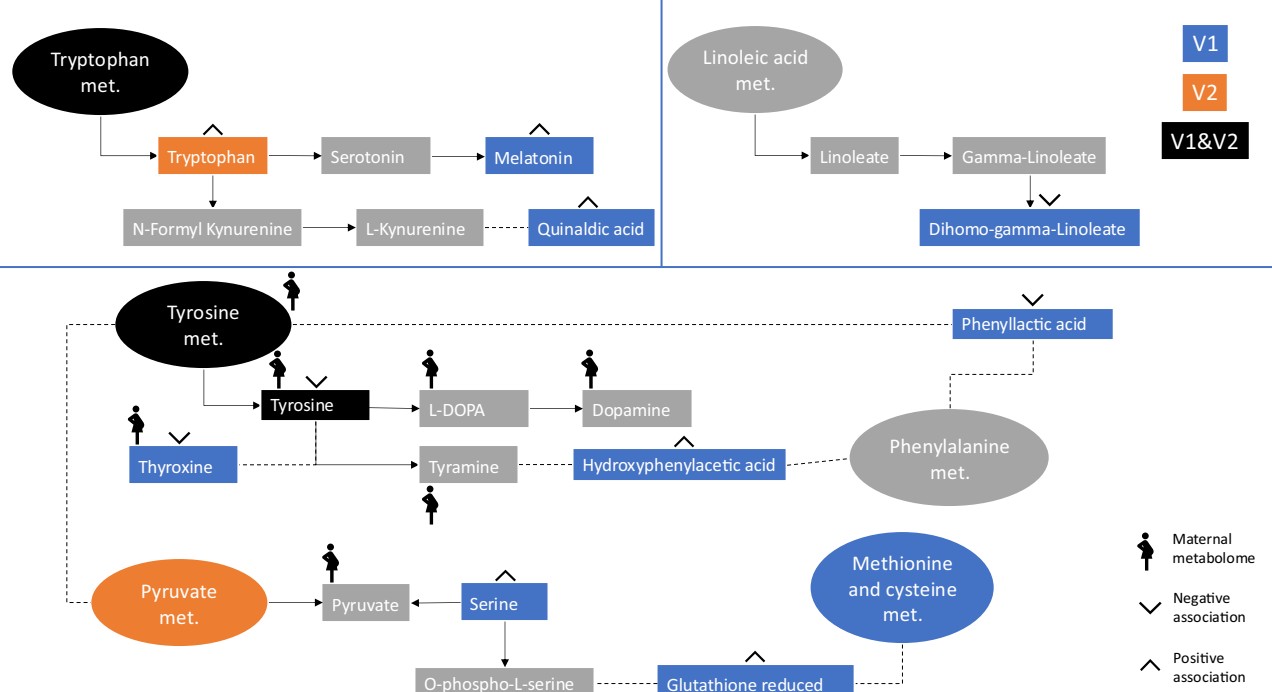

**Fig. 2 | Potential molecular mechanisms underlying the effects of prenatal phthalate exposure on the newborn metabolome from the metabolomic-wide association study (visit 1 $n$ = 216; visit 2 $n$ = 145).** Items in circles denote pathways and items in rectangle boxes denote metabolites. Blue pathways and metabolites were found in visit 1 (V1). Orange pathways and metabolites were found in visit 2 (V2). Black pathways and metabolites were found in both visits. Gray pathways and metabolites are provided for biological context but were not explicitly identified in this research. Solid arrows represent metabolites directly upstream/downstream of each other. Dashed lines represent a biochemical pathway linking metabolites not shown for clarity. Up arrows represent a positive association and down arrows a negative association. Twenty metabolites were not included, as no clear biological link was identified (see Table 3). Those marked with the maternal metabolome symbol were identified in a paper looking at the metabolic perturbations from phthalates in the maternal metabolome in this same cohort[17]. Source data are provided as a Source Data file.

highlight areas for further exploration within the microbiome, offering insights into how microbial composition and activity may influence metabolic pathways. Although investigating the microbiome is beyond the scope of this research, it is crucial to consider factors such as the type of birth (vaginal or cesarean) when studying newborn populations. These factors can significantly impact the microbiome and subsequent metabolic profiles. Therefore, future studies should ensure sample sizes are sufficiently large to allow for stratification based on these variables.

This study had several notable strengths. First, this is a well-established, socioeconomically diverse African American prospective birth cohort with validated processes for exposure measurements and metabolic data collection from newborn DBS samples. As this is a prospective cohort design, we also had well-established temporality between the exposure and outcome. Next, this study leveraged both an untargeted MWAS and MITM framework to identify and measure links between phthalate exposures, metabolomics signatures, and infant health outcomes. Finally, this research is among an all-African-American population, a group that is currently underrepresented in this existing literature with a higher risk of exposure to environmental contaminants such as phthalates, compared to their white counterparts[6]. The study also had several limitations to note. This study was limited to US-born participants without active chronic medical conditions or chronic medication use, limiting the generalizability of these results to populations with these characteristics. There were also limitations to consider with the covariate set. First, there was also no information on diet available, which would be an important covariate with phthalates, as much of the human exposure is from food and drink packaging[1]. Future studies looking at phthalate exposure should collect this information. Next, there was no information available from the substance use variables on frequency,

intensity, and duration of use, information future studies could collect to prevent residual confounding. Further, as this is a pregnancy cohort, women may have underreported their use due to social desirability bias, introducing the potential for misclassification bias. However, reported use of alcohol and tobacco in this population is only slightly lower than national estimates and use of marijuana are higher than national estimates[41], so we hypothesize any effect from this bias would be minimal. Beyond covariates, the phthalate measurement was a one-time snapshot of an exposure level that is dynamic and can change over time, which could misrepresent an individual's overall exposure experience, although research indicates a one-time sample can be representative[42]. Additionally, this research only focused on exposures to individual phthalate metabolites, and as demonstrated in this study, there could be other co-exposures that may contribute to the true effect of this environmental contaminant on health. Further, a study in this cohort demonstrated the observed negative association between environmental contaminants and adverse pregnancy outcomes can strengthen when including psychosocial stressors as an additional exposure[43]. Although beyond the scope of the current work, future studies could utilize mixture methods to consider additional contaminants and psychosocial stressors, especially as this population experiences discrimination on the basis of both race and gender[38]. Additionally, this research considered each timepoint cross-sectionally. Future studies should consider more sophisticated approaches to independently assess the effects of multiple exposure time points on the metabolomic profile. However, our current approach serves as a foundational step in understanding the temporal dynamics of phthalate exposure and its impact on the newborn metabolome. Next, due to the highly multidimensional analysis, there is an increased risk for type I errors. This was addressed by imposing stringent criteria whenever possible throughout the analysis, including

**Table 4 | Chemical identification of the metabolic features associated with phthalate exposure and NICU Net Neurobehavioral Scale (NNNS) outcome in the meet-in-the-middle analysis at visit 1 (n = 81) and visit 2 (n = 71) at p < 0.05 using generalized linear models**

| m/z | rt | Ontology level* | Metabolite | Exposure | Association with exposure (β) | Outcome | Association with outcome (β) | Pathways | Biological function |
|---|---|---|---|---|---|---|---|---|---|
| **Visit 1 (8–14 weeks) n = 81** | | | | | | | | | |
| 211.0845 | 2.84 | OL2b | 2-amino-3-(4-hydroxy-3-methoxyphenyl)propanoic acid | MBP | 0.0050 | Arousal | 0.1241 | Pharmaceutical (foodome) | Not enough information |
| 176.095 | 2.07 | OL1, OL2a | Cotinine, Serotonin | MECPP, MEHP, ΣDEHP | 0.0201, 0.0503, 2.036 | Arousal | −0.3115 | Biomarker (environmental) Tryptophan metabolism (endogenous) | Cotinine is a biomarker for tobacco use which can impact the nervous system[33] Serotonin is a neurotransmitter[28] |
| 235.0878 | 5.17 | OL2b | Acetyl-S-(3-hydroxypropyl-1-methyl)-L-cysteine | MEP | −0.0002 | Arousal, Attention | −0.0873, −0.0929 | Biomarker (environmental) | Biomarker for tobacco use can impact the nervous system[33,34] |
| 173.048 | 2.38 | OL2b | Quinaldic acid | MiBP | 0.0034 | Arousal | −0.0965 | Tryptophan metabolism (endogenous) | Not enough information |
| 196.074 | 8.59 | OL2a | 3,4,5-trimethoxybenzaldehyde | MiBP | −0.0021 | Arousal | 0.0655 | Pharmaceutical (foodome) | Not enough information |
| 416.329 | 15.04 | OL2b | Calcitriol | MBP | −0.0021 | Attention | −0.0484 | Steroid biosynthesis (endogenous) | Increases the amount of calcium in the body[83] |
| 172.146 | 12.59 | OL2a | Decanoic acid | MEP | −0.0037 | Attention | −0.7346 | Fatty acid (endogenous) | Reduces oxidative stress in neuroblastoma cells[84] |
| 194.08 | 5.94 | OL1 | Caffeine | MEP | 0.0001 | Attention | 0.0460 | Caffeine metabolism (drug) | Not enough information |
| 355.1936 | 15.44 | OL2b | JWH-019 | MEP | −0.0004 | Attention | −0.1770 | Biomarker (environmental) | Synthetic cannabinoid[85] |
| 284.076 | 3.23 | OL1 | Xanthosine | MiBP | −0.0030 | Attention | 0.0736 | Purine metabolism (endogenous) | Involved in the synthesis and degradation of nucleic acids during rapid DNA/RNA turnover[86] |
| 278.1518 | 14.36 | OL2b | Monoethylhexyl phthalic acid, Mono-n-octyl phthalate | MiBP | −0.0050 | Attention | −0.1130 | Biomarkers (environmental) | Phthalate exposure metabolites MEHP and MnOP[87,88] |
| 186.1256 | 10.1 | OL2a | 10-hydroxy-2-decenoic acid | MiBP | 0.0019 | Attention | 0.0486 | Fatty acid (endogenous) | Not enough information |
| **Visit 2 (24–30 weeks) n = 71** | | | | | | | | | |
| 105.043 | 0.57 | OL2b | Serine | MEHP, MiBP | −0.0222, −0.0051 | Attention | 0.0649 | Glycine, serine and threonine metabolism (endogenous) | Neurotransmitter precursor[23] |
| 165.079 | 3.31 | OL2a | Phenylalanine | MEHP | −0.0209 | Attention | −0.0228 | Phenylalanine metabolism (endogenous) | Found in high levels in breastmilk and is a neurotransmitter precursor[30] |
| 173.105 | 6.67 | OL2b | Acetylleucine, Hexanoyl Glycine | MEHHP, ΣDEHP | 0.0209, 1.9220 | Arousal | −0.2026 | Valine, leucine and isoleucine metabolism (endogenous) | Decreased neural inflammation in mouse models[81] |
| 176.095 | 2.56 | OL1, OL2a | Cotinine, Serotonin | MiBP | −0.0188 | Arousal | −0.4044 | Biomarker (environmental) Tryptophan metabolism (endogenous) | Cotinine is a biomarker for tobacco use which can impact the nervous system[33] Serotonin is a neurotransmitter[28] |
| 179.0946 | 10.72 | OL2a | Propham | MiBP | 0.0022 | Attention | −0.0519 | Pesticide (environmental) | Not enough information |
| 180.0786 | 10.35 | OL2a | 4-hydroxy-3-methoxycinnamic alcohol | MiBP | 0.0022 | Attention | −0.0519 | Metabolite from fruits and vegetables (foodome) | Beneficial to gut microbiota and lipid metabolism[89] |

**Table 4 (continued) | Chemical identification of the metabolic features associated with phthalate exposure and NICU Net Neurobehavioral Scale (NNNS) outcome in the meet-in-the-middle analysis at visit 1 (n = 81) and visit 2 (n = 71) at p < 0.05 using generalized linear models**

| m/z | rt | Ontology level* | Metabolite | Exposure | Association with exposure (β) | Outcome | Association with outcome (β) | Pathways | Biological function |
|---|---|---|---|---|---|---|---|---|---|
| 181.074 | 1.56 | OL2a | Tyrosine | MEP | −0.0002 | Attention | −0.0608 | Tyrosine metabolism (endogenous) | Thyroid hormone; low level associated with increase illness and neurodevelopmental issues in newborns[25] |
| 183.0895 | 0.77 | OL2b | Epinephrine | MBP | 0.0013 | Attention | −0.0427 | Tyrosine metabolism (endogenous) | Hormone and neurotransmitter[90] |
| 186.1256 | 10.1 | OL2a | 10-hydroxy-2-decenoic acid | MBP | −0.0021 | Attention | 0.0629 | Fatty acid (endogenous) | Not enough information |
| 193.0739 | 7.21 | OL2a | 3-Methylhippuric acid | MEP | 0.0003 | Arousal | −0.1245 | Urinary metabolite (environmental) | A normal urine metabolite increased by consumption of phenolic compounds[91] |
| 220.085 | 2.72 | OL2a | 5-Hydrotryptophan | MBP | −0.0017 | Arousal | −0.0543 | Tryptophan metabolism (endogenous) | Serotonin precursor[29] |
| 229.0463 | 5.43 | OL1 | Flonicamid | MBP, MECPP, MEHHP, MEOHP, MEHP, MiBP, ΣDEHP | 0.0051, 0.0171, 0.0210, 0.0277, 0.0343, 0.0077, 2.0323 | Arousal | −0.1524 | Pesticide (environmental) | Not enough information |
| 233.072 | 4.58 | OL2a | (R)-N-Acetyl-S-(1-(hydroxymethyl)-2-propen-1-yl)-L-cysteine | MECPP, MEHHP, MEOHP, ΣDEHP | −0.0105, −0.0124, −0.0177, −1.1830 | Arousal | −0.1781 | Acrylamide metabolite (environmental) | Acrylamide is classified as a possible human carcinogen; found in cigarettes or processed foods[32] |
| 330.2559 | 16.54 | OL2a | Docosapentaenoic acid | MEP | −0.0004 | Arousal | 0.1572 | Fatty acid (endogenous) | Anti-inflammatory; Found in high concentrations in breastmilk[74] |
| 376.2977 | 15.66 | OL2a | Lithocholic acid | MBP | 0.0036 | Attention | −0.1203 | Bile acid biosynthesis (endogenous) | Displays anti-neoplastic and anti-carcinogenic properties[92] |
| 400.3341 | 16.1 | OL2b | Calcifediol, 7-Ketocholesterol | MBzP | 0.0297 | Attention | 0.0725 | Steroid biosynthesis (endogenous) | Increases the amount of calcium in the body[83] Oxidation product of cholesterol[93] |
| 406.2719 | 13.08 | OL1 | 7 & 12-Ketochenodeoxycholic acid | MBP, MiBP | −0.0101, −0.0038 | Attention | 0.4866 | Bile acid biosynthesis (endogenous) | Not enough information |
| 408.2876 | 14.13 | OL2b | Cholic acid | MBP, MEHP, MiBP | 0.0011, 0.0067, 0.0015 | Attention | −0.0228 | Bile acid biosynthesis (endogenous) | Not enough information |

*MEP* monoethyl phthalate, *MBP* mono-n-butyl phthalate, *MiBP* monoisobutyl phthalate, *MBzP* monobenzyl phthalate, *MEHP* mono-2-ethylhexyl phthalate, *MEOHP* mono-2-ethyl-5-oxohexyl phthalate, *MEHHP* mono-2-ethyl-5-hydroxyhexyl phthalate, *MECPP* mono-2-ethly-5-carboxypentyl phthalate, *ΣDEHP* sum of Di-2-ethylhexyl phthalate metabolites.

*Ontology level indicates the strength of the evidence supporting each metabolite's identification or annotation, considering retention time, exact mass (MS), MS/MS fragmentation pattern, and isotopic ion pattern. Metabolites that closely matched the IESL by RT (within ±0.5 min), MS (within <5 ppm), and displayed a high degree of similarity in MS/MS fragmentation patterns (similarity score >30) received an OL1 label. For metabolites matching by RT and MS, they were labeled OL2a. An OL2b label was assigned to signals that matched the IESL by MS and MS/MS but fell outside the RT window (±0.5 min).

The chemical identity was conducted by matching peaks by accurate mass and retention time to authentic reference standards in an in-house library run under identical conditions using tandem mass spectrometry. Source data and exact two-sided p-values are provided as a Source Data file. Atlanta African American Maternal-Child cohort, 2016–2018.

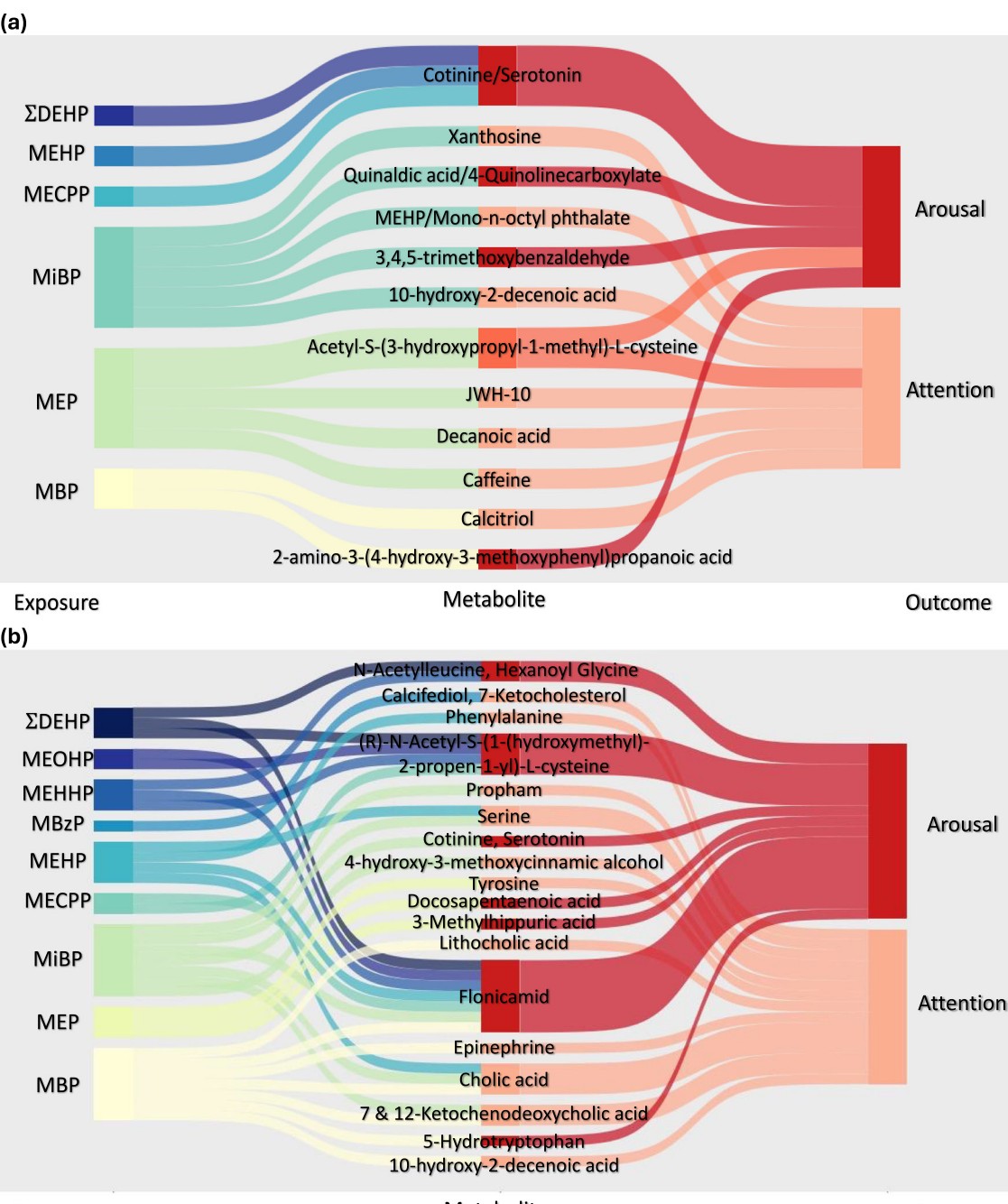

**Fig. 3 | Sankey plots illustrating associations between phthalate exposure, metabolite perturbations, and NICU Neonatal Neurobehavioral Scale (NNNS) outcomes at two visits.** Sankey plots in **a** are from visit 1 (*n* = 81), and **b** are from visit 2 (*n* = 71). The first node is the phthalate exposures, leading to the associated metabolites in the second node, and ending at the NNNS outcomes, attention and arousal, in the last node. The yellow/blue colors on the left-hand side indicate different phthalate metabolites. The red colors on the right-hand side differentiate attention (dark red) and arousal (light red) or both (orange) outcomes. Only significant, annotated findings are displayed in this figure; for details on the significance level and beta values, please see Table 4. Source data are provided as a Source Data file. MEP Monoethyl phthalate, MBP Mono-n-butyl phthalate, MiBP Monoisobutyl phthalate, MBzP Monobenzyl phthalate, MEHP Mono-2-ethylhexyl phthalate, MEOHP Mono-2-ethyl-5-oxohexyl phthalate, MEHHP Mono-2-ethyl-5-hydroxyhexyl phthalate, MECPP Mono-2-ethly-5-carboxypentyl phthalate, ΣDEHP Sum of Di-2-ethylhexyl phthalate metabolites.

applying the Benjamini-Hochberg FDR. Unfortunately, for the MWAS pathway analysis and the entirety of the MITM analysis, we could not use FDR-corrected significance thresholds due to the limited statistical power resulting from the sample size, as the more stringent threshold would not include a sufficient number of metabolic features to conduct a meaningful pathway analysis or have MITM results. Therefore, these results should be carefully interpreted and confirmed using additional studies with larger sample sizes. The use of less invasive

techniques for metabolomics studies, such as newborn DBS, could facilitate larger cohorts. Regarding newborn DBS samples obtained through heel-sticks, issues have been pointed out, such as uneven cell distribution, cell lysis, hematocrit variation, and pre-analytical clotting when employing untargeted metabolomics analysis on this biomatrix[44-46]. Despite the normalization of untargeted DBS metabolomics analysis based on the size and weight of the DBS, limitations in sample volume prevented the application of additional, potentially

more optimal normalization methods, such as adjustments for hemoglobin, specific gravity, protein, or potassium[47]. Further, as this investigation used a single, 15-mm punch, this sample may not represent larger portions of blood because of potential variation in signals dependent on spotting and drying time. Using NNNS scores as an outcome presents certain considerations. The design of NNNS is intended to encompass a broad spectrum of neurobehavioral domains, including attention and arousal. While this lack of specificity might limit its precision in pinpointing specific clinical phenomena, it enables the test to capture general trends in infants. This broader scope facilitates comparability across infants within a study population[48]. Although NNNS scores may not provide clinical recognition of a specific condition, their ability to highlight general trends offers valuable insights into both present and future neurobehavioral function[49-51]. Finally, there are fewer assumptions required for using the MITM framework compared to formal mediation analysis, and the results are suggestive of future hypothesis testing in other samples and should be interpreted with caution.

Taken together, this research supports the current mechanistic understanding of how prenatal phthalate exposure influences the newborn metabolome and how these biological perturbations might impact neurobehavioral outcomes. Additional research is needed to evaluate the generalizability of these findings, test for differences between African Americans and other populations, incorporate additional co-exposures, and investigate the potential of these metabolic responses for detection, treatment, and prevention strategies in public health and clinical settings.

## Methods

### Study population

The project utilizes data from the Atlanta African American Maternal-Child Cohort, a prospective cohort that recruited pregnant African American individuals from Grady Memorial Hospital and Emory University Hospital Midtown since 2014[52]. The study was approved by the Emory University Institutional Review Board, and all participants provided written informed consent prior to enrollment.

Participants were eligible for the study if they self-reported non-Hispanic African American or Black race, were US-born, 18–40 years old, and were pregnant with a singleton fetus between 8 and 14 weeks gestation. As the primary goal of this cohort was to identify what may be contributing to adverse birth outcomes in African Americans in otherwise healthy women, participants were excluded if there was any active diagnosis of chronic medical conditions or chronic use of prescription medications (verified through the prenatal medical record). The study participants included in the metabolome-wide association study (MWAS) were further restricted to mother-newborn pairs with mothers who had urine samples available for the determination of urinary phthalate levels and newborns with metabolomics profiles from dried blood spot (DBS) samples (visit 1 $n = 216$; visit 2 $n = 145$; Fig. S1). Participants could be in visit 1 only ($n = 76$), visit 2 only ($n = 5$), or represented in both visits ($n = 140$). The MITM analysis was further subset to those mother-newborn pairs for whom NNNS data was available (visit 1 $n = 81$; visit 2 $n = 71$). For the cohort, prenatal data was collected during two clinical visits: the first collection was targeted to occur during the 8–14 week gestational period and the second during the 24–30 week gestational period. During the first visit in early pregnancy, sociodemographic data such as maternal age and education were collected from self-reported surveys and from prenatal administrative records review. Health information was completed by self-report, including prior pregnancies (if any) and substance use. For substance use, medical records were reviewed for reported use prior to pregnancy, and use during pregnancy was self-reported. During the first and second visits, prenatal urine samples were collected (and assayed as described below), and clinical data (such as height and weight, from which maternal body mass index and weight gain were

calculated) were collected. Additionally, after delivery, medical record abstraction was used to ascertain the infant's gestational age at birth, weight, and sex. Covariate information used in this analysis was pulled from these data sources.

### Exposure collection

Previous research has demonstrated that a one-time urine sample can capture a representative picture of recent phthalate exposure, even with the relatively short half-life (around 12 h)[42] of these compounds. Metabolites of eight phthalates were measured in urine collected during the two prenatal clinical visits (Table S1). These phthalate metabolites were: monoethyl phthalate (MEP), mono-n-butyl phthalate (MBP), monoisobutyl phthalate (MiBP), monobenzyl phthalate (MBzP), mono-2-ethylhexyl phthalate (MEHP), mono-2-ethyl-5-oxohexyl phthalate (MEOHP), mono-2-ethyl-5-hydroxyhexyl phthalate (MEHHP), and mono-2-ethyl-5-carboxypentyl phthalate (MECPP). Briefly, urine samples (0.5 ml) were mixed with stable isotopically labeled analogs of the phthalate metabolites and 2000 units of β-glucuronidase in 1 mM ammonium acetate (pH 5) buffer and incubated overnight at 37 °C[13]. Then, enzyme activity was terminated by adding 0.15 M sodium phosphate buffer to the sample. Phthalate metabolites were extracted using the Bond Elut solid phase extraction cartridge (Agilent Technologies, Inc., Santa Clara, CA). These samples were reconstituted with Milli-Q water, and the target compounds were separated using high-performance liquid chromatography on a Betasil Phenyl (3μ,150 × 2.1 mm) (Thermo Scientific, San Jose, CA) analytical column and finally were analyzed by tandem mass spectrometry on an Agilent 6460 triple quadrupole mass spectrometer (Agilent Technologies, Inc). Concentrations were derived from a regression analysis of the peak area of the analyte ion divided by the peak area of the internal standard ion and the calibrant concentrations. Quality assurance and control procedures were also incorporated into the analyses (i.e., analyses of NIST reference materials and quality control samples). The analytical method was certified by successful participation in the German External Quality Assessment Scheme.

As MEHP, MEHHP, MEOHP, and MECPP are metabolites of the same phthalate exposure, DEHP, these exposures were considered together as the molar sum of these analytes (ΣDEHP). The molar sum of DEHP metabolites was calculated using the following formula[17,53]:

$$\sum DEHP = MEHP/278.344 + MEHHP/294.34$$
$$+ MEOHP/292.33 + MECPP/308.33$$

As previously described, the limits of quantification (ng/mL) for these phthalates metabolites were: 1.0 (MEP), 4.0 (MBP), 2.0 (MiBP), 0.2 (MBzP), 0.2 (MEHP), 0.4 (MEOHP), 0.4 (MEHHP), and 5.25 (MECPP)[17]. The linear range was 0.01–50 ng/mL for all phthalate metabolites except for MEP, which was 5–500 ng/mL. Samples that fell above those concentrations were diluted and repeated. Accuracy based upon the National Institute of Standards and Technology (NIST) was 91–105%. The percent relative standard deviations were all below 12%. Any values that fell below the LOD were assigned the LOD divided by the square root of 2 ($LOD/\sqrt{2}$)[54,55]. To account for the variability of urine dilution, all exposure variables were creatinine adjusted by adding it as an independent covariate in the final model[56,57].

### Metabolomic data collection

This study used newborn DBS samples, which are routinely collected shortly after delivery for medical screening and public health surveillance[58-61]. Previous research has demonstrated that DBS samples are representative biospecimens of the human metabolome and are comparable to whole blood samples containing serum and plasma while having the added advantage of being less invasive and allowing for insights into the early metabolomic response to prenatal exposures, which are largely understudied[61]. DBS samples were collected

via heel stick within 24–48 h of birth. This procedure followed a standardized protocol, including the disinfection of the skin, sample collection using a 2.5 mm lancet, and obtaining ~75 μL of blood on a Guthrie card[62]. Samples were stored in a walk-in refrigerator (2–8 °C) for up to three months until they were transferred to Emory University and stored within gas-impermeable bags with dessicant in a freezer (−80 °C) until further processing. The site used consistency in the collection, storage, and shipment of biospecimens, and any signals that quickly degrade or would have variability in degradation due to sample storage time would not be expected to arise as significantly associated with the studied phenotype. In this cohort, single 15-mm punches (~50 μL of whole blood) were received from the Guthrie cards between 2016 and 2020[59].

Untargeted high-resolution metabolomics profiling was performed in a single run at the North Carolina HHEAR Hub, located at the UNC Nutrition Research Institute within the North Carolina Research Campus (Kannapolis, NC), using established methodologies[58,63,64]. Briefly, DBS samples underwent extraction with 1 mL ice-cold methanol, including 500 ng/mL L-tryptophan-d5. The samples were vortexed at 5000 rpm for 10 min and sonicated for an additional 20 min. After centrifugation at $16,000 \times g$ for 10 min at 4 °C, 600 μL of the extracted supernatant was transferred into a 2.0 mL LoBind microfuge tube. A combined 70 μL of the extracted supernatant from all study samples was pooled and then redistributed into new tubes, each containing 600 μL, to generate the total study pool samples. Empty blood spot cards (without blood), matching the size and material of study samples, were used as blanks and processed using identical procedures. The supernatant (70 μL) from each of the blanks was mixed together and then distributed into multiple low-bind 2.0 mL tubes, each containing 600 μL. All supernatant aliquots (600 μL), including study samples, study pools, and blanks, were dried by vacuum concentrator overnight and reconstituted with 100 μL water-methanol (95:5, v/v) for ultra-high performance (UHP) LC-HR-MS analysis. External quality control samples consisted of 50 μL aliquots of NIST reference plasma (SRM 1950) and were prepared using a similar procedure. After study sample randomization, quality control study pools, NIST plasma references, and blanks were interspersed at a rate of 10% among the study samples during the analysis sequence, which was divided into three batches.

The untargeted data was obtained using a Vanquish UHPLC system coupled with a Q Exactive™ HF-X Hybrid Quadrupole-Orbitrap Mass Spectrometer (Thermo Fisher Scientific, San Jose, CA). For analysis, a 5 μL volume was injected. The HSS T3 C18 column (2.1 × 100 mm, 1.7 μm, Waters Corporation) was used to separate metabolites at 50 °C with water and methanol as mobile phases, each containing 0.1% formic acid (v/v). The UHPLC linear gradient was initiated with 2% methanol and then reached 100% methanol in 16 min and was held at this level for 4 min, with a flow rate of 400 μL/minute. The untargeted metabolomics data was acquired in the 70–1050 m/z range using the data-dependent acquisition mode for the MS/MS spectra data. Subsequently, the UHPLC-HR-MS data underwent processing using Progenesis QI (version 2.1, Waters Corporation) for peak identification and alignment. Background signals were excluded if their mean intensity across blanks exceeded that of the quality control study pools based on the unnormalized data. The remaining peaks were normalized using the "normalize to all" feature in Progenesis QI. Signals exhibiting significant differences across the three running batches (with false discovery rate (FDR) correction $q < 0.05$) were excluded from further analysis. Finally, the metabolomics data was filtered to include only those detected in 95% of serum samples and subsequently $\log_{10}$-transformed to normalize data.

## NNNS scores
We evaluated infant neurodevelopmental functioning using the NNNS[65]. Initially developed for the National Institutes of Health

(NIH) to evaluate drug-exposed and at-risk infant, the NNNS is a validated measure of early neurodevelopment in very young infants[65]. The measure has demonstrated validity across multiple studies, including those with US, and Black participants[50,66]. The complete assessment was designed to provide comprehensive, theoretically driven, and psychometrically sound measures in 12 subdomains, including attention (i.e., information processing) and arousal (i.e., state control and excitability). Although originally developed for infants prenatally exposed to drugs of abuse, previous research has demonstrated that the scores can be applied to low-risk infants with no known drug exposure[49]. Further, research has indicated that NNNS scores are associated with differences in behavior in early childhood[49–51]. In this study, the NNNS was administered when infants were on average 2–4 weeks old (corrected for gestational age a birth) by trained and certified research specialists among those whose families agreed to participate in the infant follow-up. The NNNS administration did not vary across the 2–4-week age range. The NNNS was administered in a quiet location in the participant's home or the lab setting, according to standardized procedures. The measure was administered by one research specialist, with a second research specialist present at the visit, filming the administration for reliability checks. Both examiners were blind to the newborn's exposure group. All examiners completed a 5-day training program and established adequate inter-rater reliability with a NNNS trainer. Caregivers were present in the home or the lab setting but did not participate in the administration of the NNNS.

NNNS standardized arousal and attention summary scores were created on the basis of confirmatory principal components analyses and were used in the statistical analyses. The first composite score assesses newborn arousal by combining summary scores related to excitability, arousal, the number of calming strategies required by the examiner, physiological signs of stress (e.g., tremors, startles), and the newborn's self-regulation capacity (reverse scored). Higher scores on this composite indicate increased levels of newborn arousal. The second composite score measures the newborn's attention, reflecting their ability to respond to, focus on, and track external stimuli. This score is summarized by components related to attention and lethargy (reverse scored), with higher scores indicating more mature levels of newborn attention. In this study, standardized scores were created from the averages of each component. Previously published research has detailed the correlations between NNNS summary scores for this sample and demonstrated that the summary scores in each composite were moderate to highly correlated, supporting the creation of two overarching scores[67].

## Statistical analyses
We analyzed phthalate metabolite data from the two study prenatal visits cross-sectionally. The use of two separate regression models allows us to examine the potential differential impacts of phthalate exposure during early and late pregnancy, thereby offering preliminary indications of critical periods without the added complexity of a longitudinal analysis that might compromise the integrity of our analysis. A descriptive analysis of selected covariates was conducted with a summary of continuous variables presented using the median and interquartile range (IQR) and categorical variables presented as the count ($n$) and frequency (%).

For the untargeted MWAS, generalized linear models were applied to examine the association between phthalate metabolite concentrations in early pregnancy (visit 1) and in late pregnancy (visit 2) and newborn metabolic feature intensities. All models controlled for mother's age, education, parity, alcohol consumption, tobacco use, marijuana use, pre-pregnancy BMI, urine creatinine levels, as well as the baby's sex and gestational age (GA) in weeks at the time of the

study visit. The model took the following form:

$$\log_{10}\left(Y_{ij}\right) = \alpha + \beta_{1j}(\text{Phthalate}_i) + \gamma_{1j}(\text{age}_i) + \gamma_{2j}(\text{education}_i)$$
$$+ \gamma_{3j}(\text{parity}_i) + \gamma_{4j}(\text{alcohol use}_i) + \gamma_{5j}(\text{tobacco use}_i)$$
$$+ \gamma_{6j}(\text{marijuana use}_i) + \gamma_{7j}(\text{BMI}_i) + \gamma_{8j}(\text{creatinine}_i)$$
$$+ \gamma_{9j}(\text{sex}_i) + \gamma_{10j}(\text{GA}_i) + \epsilon_{ij}$$

Where $\log_{10}(Y_{ij})$ refers to the normalized (log base 10) intensity of the newborn metabolic feature $j$ for participant $i$, $\alpha$ is the intercept, and $\beta_{1j}(\text{Phthalate}_i)$ is one of the nine different phthalate metabolite exposures measured in this study (MEP, MBP, MiBP, MBzP, MEHP, MEOHP, MEHHP, MECPP, and ΣDEHP). $\epsilon_{ij}$ represents random error. Other covariates were selected through a thorough review of the literature and a directed acyclic graph (Figure S2) to identify covariates that could potentially introduce confounding bias[68]. The $p$ values obtained from the MWAS were corrected for FDR using the Benjamini-Hochberg procedure since multiple comparisons were made[69].

### Pathway enrichment analysis

A metabolic pathways enrichment analysis was conducted in the full MWAS to predict the functional biological activity of uncharacterized metabolic features that were associated with at least one urinary phthalate level. Due to a limited number of significant features at the FDR-corrected $q < 0.2$ threshold (Table S4), the less stringent cutoff of $p < 0.05$ was used for pathway analyses. All pathway analyses were conducted in Python (version 3.9) using *mummichog* (version 2.3)[19], a bioinformatics tool that is able to utilize the mass spectrometry data from unidentified metabolic features to infer biological pathways. The program further provides an adjusted $p$ value for each pathway by resampling the reference input file, assuming a gamma distribution. To minimize the chance of false positive discoveries, only pathways that had an adjusted $p < 0.05$ after 1000 permutations were performed, and pathways enriched with ≥10% overlapping signals relative to the pathway size were retained.

### Chemical identification and annotation

To avoid false positive matches, only metabolites that met the FDR-corrected $q < 0.2$ threshold were annotated. Metabolomic signals were identified by comparing them to the NC HHEAR Hub's IESL, encompassing over 2400 compounds. This library contains both endogenous and exogenous metabolites. The endogenous metabolites are linked to an individual's metabolism. In contrast, exogenous metabolites are associated with lifetime exposures, such as environmental contaminants, dietary compounds, pharmaceuticals, and secondary metabolites formed after exposure. All standard reference compounds in the IESL were analyzed under identical conditions as the study samples. The same signals were cross-referenced with publicly available databases, including NIST and METLIN, for further verification[58].

The matching process involved assigning ontology levels (OL) to signals, indicating the strength of evidence supporting their identification or annotation. These levels considered various factors, including retention time (RT), exact mass (MS), MS/MS fragmentation pattern, and isotopic ion pattern. Metabolites that closely matched the IESL by RT (within ±0.5 min), MS (within <5 ppm), and displayed a high degree of similarity in MS/MS fragmentation patterns (similarity score >30) received an OL1 label. For metabolites matching by RT and MS, they were labeled OL2a. Metabolites that earned an OL1 or OL2a label were considered confidently identified in this study. An OL2b label was assigned to signals that matched the IESL by MS and MS/MS but fell outside the RT window (±0.5 min). These metabolites were often conjugates or isomers that shared similarities with the matched compound in the IESL. Signals matching with MS (<5 ppm) and exhibiting a strong resemblance in experimental MS/MS (similarity score >30) to

public databases received a public database (PDa) label. Metabolites confirmed with OL2b and PDa labels were considered as confidently annotated in this study, while those with less confidence were not included in subsequent analyses.

### Meet-in-the-middle analysis

The MITM approach has been used by previous research to search for early biological effects and potential intermediate biomarkers in studies of environmental contaminants and reproductive health[58,70,71]. First, the population used in the larger MWAS was a subset of those with the NNNS score outcome available (Visit 1 $n = 81$; Visit 2 $n = 71$). MWAS were run to determine which of the newborn DBS signals were associated with prenatal phthalate exposure and NNNS score. First, the exposure-metabolomic relationships were determined for each urinary phthalate metabolite using a linear model controlling for the same covariate set as the larger MWAS model. A linear model was then run to investigate the metabolomic signal-outcome relationship for NNNS scores, controlling for the same covariate set, except for creatinine, as there were no urinary measures in the model. Chemical identification followed the same procedural steps as outlined in the larger MWAS; however, due to the limited sample size, unadjusted $p$ values ($p < 0.05$) were used in this exploratory MITM analysis. The metabolites that were identified are indicators of a potential intermediate mechanism that could be explored further in future research.

All analyses were conducted using R (Version 4.3.0, https://www.R-projects.org) or Python (Version 3.9, https://www.python.org).

### Reporting summary

Further information on research design is available in the Nature Portfolio Reporting Summary linked to this article.

## Data availability

The raw and processed metabolomics data from this study have been archived in the Metabolomics Workbench (https://www.metabolomicsworkbench.org/, Study ID ST002692) through the UNC HHEAR Laboratory. Due to privacy considerations, outcome and PFAS exposure data are accessible only through restricted access; requests can be directed to the corresponding author, Dr. Liang, and will be addressed within 10 business days. Demographic covariate data are not publicly available due to data privacy regulations. Source data for figures and tables, along with coding materials and data protocols, are provided in the Supplementary Information/Source Data file. Source data are provided with this paper.

## Code availability

Specific code is available from the corresponding author, Dr. Liang, and the request will be responded to within 10 business days.

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

## Acknowledgements

We thank all members of the Environmental Metabolomics and Exposomics Research Group at Emory (EMERGE) for their valuable input and feedback on this project. S.S.H. and K.R.T. are supported by the NIEHS T32 Training Program in Environmental Health and Toxicology (T32 ES012870). This work was supported by the NIH research grants [R01NR014800, R01MD009064, R24ES029490, R01MD009746, R21ES032117, R01ES035738], NIH Center Grants [P50ES02607, P30ES019776, UH3OD023318, U2CES026560], and Environmental Protection Agency (USEPA) center grant [83615301].

## Author contributions

S.S.H. completed the analysis, interpretation, and writing of the manuscript. D.L., S.S.H., and T.H. generated the study aims and prepared the manuscript. Z.T. completed the code checks, manuscript editing, and preparation. A.D., P.A.B., and E.J.C. were involved in data collection, cohort management, and manuscript editing and preparation. B.R., S.L.M., and S.S. were involved in metabolomics data processing, manuscript editing, and preparation. T.H., S.M.E., K.R.T., Y.T., P.P., G.E.L., J.E., P.B.R., D.B.B., and D.P.J. were involved in manuscript editing and preparation.

## Competing interests

The authors declare no competing interests.
