## [Transparent Peer Review file · Nature Communications]

Impact of Prenatal Phthalate Exposure on Newborn Metabolome and Infant Neurodevelopment

Corresponding Author: Dr Donghai Liang

Version 0:

Reviewer comments:

Reviewer #1

(Remarks to the Author)

My expertise is infant neurobehavior so I will confine my comments accordingly:

Line 200: reference 32 is incorrect for this statement and the NNNS was not developed by the NIH it was developed for the NIH.

Line 201, reference 33 is not the correct reference for the NNNS. It was published as a supplement to Pediatrics by Lester and Tronick.

Line 209, how does reference 35 relate to NNNS?

PCA was used to derive attention and arousal scores. However, no information is presented about the PCA (percent variance accounted for, eigenvalues, item loadings etc)

Figure 3 (a and b), how do we know what are the statistically significant effects and what do the colors mean? It looks as if the arousal associations are stronger than the attention findings in both a and b.

Lines 443-450: This section is a misunderstanding of the NNNS. It is, in fact, a "highly specific functional assessment" as it takes the entire neurobehavioral repertoire of the neonate and breaks it down into 12 functional domains. The NNNS has been related to chemicals, toxins and pollutants in a number of studies. And the NNNS does have clinical significance. It has been shown to predict developmental outcome in 17 studies.

It is hard to understand what are the exact NNNS findings in this study. This seems to be a missed opportunity because one of the strengths of this study could be that because the NNNS does predict developmental outcome these NNNS findings could suggest that phthalates could have long term developmental consequences for these infants and some of the potential metabolic pathways that are responsible.

Reviewer #2

(Remarks to the Author)

Hoffman and colleagues investigated the potential mediating role of metabolomic profiles at birth in the relationship between prenatal phthalate exposure and neonatal development. The manuscript is generally well-written and provides valuable insights into the relationship between prenatal urinary phthalate levels in relation to metabolomic features measured in serum obtained from cord blood. Furthermore, it sheds light on potential links between metabolomic features and early neurodevelopment. The focus on a vulnerable and understudied population is an important strength, as is the collection of samples at two discrete periods during pregnancy and measurement of several phthalate metabolites. Complementary statistical approaches are used to arrive at the associations and plausible biological pathways are explored. However, some questions regarding the methodology and interpretation of results require further attention, as detailed below.

Specific comments/questions:

Introduction:

Line 82: The phrase "impact maternal serum samples" is not very clear. What about maternal serum samples was studied?

Line 87: Rephrasing as "results indicated" or "authors found" would be more precise.

Line 91: The mechanisms under investigation don't strictly underpin exposure but its health effects.

Lines 95-102: There are no objectives dedicated to understanding the association between phthalate exposure and NNNS scores. Has this been investigated/demonstrated in another study? If so, could this prior work be cited? If not, can the authors explain why this analysis is not being performed?

Methods:

Line 110: What is MWAS? Metabolome Wide Association Study? Spell out on first use.

Lines 107-12: Inclusion/exclusion criteria should be reported in more detail. Can you elaborate specifically on what chronic medical conditions and what medications were used as an exclusion during recruitment? It seems like there were a lot of restrictions at recruitment, which affects generalizability and selection bias, especially with the increased prevalence of chronic medical conditions in AA/Black communities.

Do the authors have any information on ethnicity (i.e., Hispanic/Latina Black women)?

Lines 124-125: The authors stated "Previous research has demonstrated that a one-time urine sample can capture a representative picture of recent phthalate exposure, even with the relatively short half-life (around 12 hours)" – This may be an overstatement, I think there are still limitations due to potential variations in exposure levels throughout pregnancy. Levels can vary based on changes in dietary/supplemental intake and lifestyle changes due to pregnancy. It would be good to mention this as a limitation in the Discussion.

Line 130: Check spelling of MECCP.

Lines 161-2: What is the stability of the metabolomic signatures at 2-8C and in the 0-3 month range that represents the storage conditions prior to long-term storage at -80C?

Line 164: Is there information available from the researchers or prior literature to understand how well data from a single 15-mm punch represent the measured features with respect to an entire blood spot?

Line 199: In this section, could the authors please indicate how many infants were assessed with the NNNS? If the number of infants is smaller at age of NNNS assessment than at exposure assessment, could the authors summarize the main reasons why NNNS could not be administered at the appropriate age?

Line 202: Could the authors indicate what other developmental measures the NNNS correlates with at later ages? In other words, how reliable is the NNNS in predicting developmental assessment scores in later infancy?

Lines 207-8: Please provide more information on the assessment and its circumstances. Where was NNNS administered and in what conditions? How many research specialists were involved in administering it? Were caregivers present and what was their role?

Also, please provide more information on the NNNS domains, what they measure and how they are assessed. Perhaps the latter could be provided as online supplemental material. Finally, newborns change developmentally between 2 and 4 weeks of age. Could the authors state whether the entire assessment is administered regardless of age or are there age-specific start points?

Statistical analysis:

Please mention the number (%) of women lost to follow-up and excluded from the analysis due to missing data. In fact, it would be useful if the authors provided a flow diagram for the selection of participants into the study.

It is unclear whether the authors assessed for potential selection bias. Were characteristics of excluded women similar to those included?

The authors mentioned using a DAG to identify confounders. I recommend adding this DAG as a figure.

Covariates were retrieved from medical records, but it is not too clear what questions were asked, as well as the time frame. Especially questions about tobacco, marijuana, and alcohol use – Was this assessed during pregnancy? Or past use? Because this is a self-report and there is no information on frequency, intensity, and duration of use, there is definitely room for residual confounding and misclassification. Further, because of social desirability, women may have underreported their use – please mention these limitations in this discussion.

Line 212: Could the authors provide some justification for this choice when exposure metrics are available at two points? Given interest in critical windows, two separate regression models weaken any conclusions to that could be made to that effect. Furthermore, when the metabolomic profile is only measured at one time following both exposures, some thought needs to be given to the fact that the levels/profiles of the features may not related to the two exposures independently.

Line 235: Could the authors give some indication of how many features were significant at the FDR-corrected $q < 0.2$ threshold and what implications the less stringent p-value cutoff had for the pathway analysis? How did loosening the cutoff affect number of features included in the study?

Were these metabolite groups analyzed separately to account for the level of similarity/certainty in identification?

Line 269: Review text.

Line 277: This is the first time that the exploratory nature of the MITM is mentioned. Could the authors clarify this in the abstract as well?

Results

Lines 293-4: Could the authors provide levels of correlation among phthalate metabolites at each time and across time?

Line 310: Only attention and arousal are mentioned as outcomes yet the NNNS was stated to yield 12 sub-domains (lines 203-4). It is not clear from the methods that attention and arousal will be the main endpoints under study. Could the authors clarify this in the methods section and provide some justification for this choice?

Table 1: how do the study samples compare on key sociodemographic characteristics (and exposure markers) to those who are excluded from analysis?

Discussion:

Line 353: There needs to be some caution regarding this conclusion b/c the analyses from visit 1 and visit 2 do not include the same set of participants, and furthermore, the number of participants was much reduced from MWAS to the MITM analysis.

Lines 388-90: This is an interesting point but brings up the question of what the range of tyrosine/tryptophan values observed in this study represents. Can they be viewed as being within a normative range? Furthermore, at what ages in references 49 & 50 were the clinical correlates of tyrosine metabolic pathways observed? Clinical disease is likely associated with values outside of a typical range and it is unclear how strongly the attention scores on NNNS correlate with developmental measures in later childhood.

Another strength of this study is the temporality between exposure and outcome due to the prospective cohort design.

One of the criteria for eligibility was US born Black women, which limits generalizability to non-US born Black women – it would be important to discuss this as a limitation.

Reviewer #3

(Remarks to the Author)

Reviewer #4

(Remarks to the Author)

Revision: Prenatal urinary phthalate metabolite levels, perturbation of the newborn metabolome, and infant neurodevelopment in the Atlanta African American Maternal Child Cohort.

The present study evaluates the association between urinary phthalates in African American pregnant women and perturbations in the blood metabolome of newborns. Furthermore, the Authors also evaluated the association between prenatal phthalate exposure and infant neurodevelopment (assessed by the NNNS score) through perturbation in the newborn metabolome. The study is based on an adjusted generalized linear model (MWAS) and a meet-in-the-middle (MITM) approach. The methodology is robust and well described.

Authors propose that phthalates determine an alteration of the metabolites involved especially in the tryptophan and tyrosine pathways, in turn responsible for the impaired neurodevelopment of the newborn. These conclusions are relevant as they may explain how phthalates affect the newborn metabolome and impact the neurobehavioral outcomes. We agree that further studies are needed to know how these effects can be generalized, and to test the differences between African Americans and other populations, the effects of other co-exposures, and the impact in public health and clinic.

The manuscript may be accepted after revision.

Main revisions:

- Line 175: The authors should better explain the population enrolled in the study. It is not clear to the reader how many women are involved in the first and/or in the second visit. In the abstract, it is reported that 278 mother-newborn pairs have

been under investigation, however only 216 participants were reported for visit 1 and 145 for visit 2. I suppose all 145 participants to the 2 visit were attending the first visit. Or not?

- Line 324-325 and Table S3: Exposure to phthalates is very similar between the first and second visits, as shown in table 2. How do you explain such a higher number of altered metabolites related to visit 1? In the discussion, the Authors proposed the importance of timing in exposure, however, the hypothesis should be better explained, by proposing possible mechanisms that protect or affect the newborns' metabolism. Specifically, this aspect should be explored for phthalates such as MEHP, MECPP, and MEOHP which appear to be correlated only in one of the two visits, despite having been measured in both. Likewise, it should be better evaluated why there is such a variable number of metabolic features associated with MBP (as reported in Table S3), especially given its relevance in the context.
- Table 3, Table 4, and discussion: Various metabolites belonging to the exposome have been correlated with phthalates. If these molecules are, for example, taken from the diet, the authors should propose a possible role of phthalates in the altered metabolism, absorption, or excretion of these molecules. Or not?

Minor revisions

- Line 73 DEHP: write the full name.
- Line 112 and 197: The Authors referred to serum metabolomics, which probably meant blood metabolomics. Or not? Moreover, the authors stated "participants who had urine and blood samples": Do they mean mother-newborn pairs?
- Line 116: The Authors evaluated if women under study followed some specific diet? As reported in the literature, phthalate exposure is linked to food and beverage consumption, but it has not been properly underlined in the background neither in the study population.
- Line 119-122: The Authors considered the type of delivery? Some reported metabolites, as 3-Hydroxyhippuric acid or Hydroxyphenylacetic acid are related to the microbiome, which can be impacted by the type of birth.
- Line 150: In the manuscript, the LOD of each phthalate has been reported. It would be of interest to also report the LLOQ, given that the phthalates have been quantified, the linearity ranges of the curve, and possibly the accuracy range considered and %CV.
- Line 316-317: The sentence "There were 28 perturbed metabolic pathways in visit 1 only and 12 in visit 2 only" could be changed to: "There were 28 perturbed metabolic pathways associated with phthalate measured in visit 1 and 12 in visit 2."? As it is written, it may be confusing for the reader. The Author are speaking of perturbed metabolic pathways associated with phthalates at this 2 specific time points.
- Line 370-371: Probably due to a typo, dihomogamma-linoleate was defined as an essential amino acid instead of polyunsaturated fatty acid.
- Figure 3: visit 2 (n=1)! Is n=1 correct? Maybe n=71?
- Table 2: It was not reported what the symbol "#" stands for.
- Table 3: The 3-Methylhippuric acid is a urinary metabolite, as correctly reported. Is it usual to find it in the new-born blood? It should be a xylene metabolite.
- Table 4: Cotinine is a biomarker for tobacco use. 13-14% of mothers smoke (calculated from Table 1). Only metabolites present in at least 95% of the DBS samples were taken into consideration. Then, how is it possible that Cotinine is present in the table? The mass 176.095 has been related to MECPP, MEHP and Σ DEHP in visit 1, while to MiBP in visit 2. Could you explain this?
- Table S3 and discussion: The MBP had a relevant role in metabolic perturbation (evident in Table S3) and it has been correlated in both visits. The authors should better clarify its role and possible differences from its analogues.

Reviewer #5

(Remarks to the Author)

Version 1:

Reviewer comments:

Reviewer #2

(Remarks to the Author)

The authors have addressed all my comments and suggestions. I have nothing further.

Reviewer #3

(Remarks to the Author)

Reviewer #4

(Remarks to the Author)

Authors propose that phthalates determine an alteration of the metabolites involved especially in the tryptophan and tyrosine pathways, in turn responsible for the impaired neurodevelopment of the newborn. These conclusions are relevant as they may explain how phthalates affect the newborn metabolome and impact the neurobehavioral outcomes.

The Authors answered adequately to our comments.

Further work is needed, as also suggested by the Authors, to evaluate and generalize the conclusions of the study, but the manuscript can now be considered for publication.

Reviewer #5

(Remarks to the Author)

Reviewer #6

(Remarks to the Author)

Thank you for your responses concerning the NNNS. I believe you have addressed all of the points made in the review of your manuscript.

Response to Reviewers

Title: Prenatal urinary phthalate metabolite levels, perturbation of the newborn metabolome, and infant neurodevelopment in the Atlanta African American Maternal Child Cohort

Thank you for the comprehensive review of our manuscript entitled “Prenatal urinary phthalate metabolite levels, perturbation of the newborn metabolome, and infant neurodevelopment in the Atlanta African American Maternal Child Cohort”. We greatly appreciate the insightful feedback and constructive comments received from the reviewers and have responded to each in the text below. Please note that all changes to the manuscript highlighted in this letter are in blue. A tracked changes version of the manuscript is also available for your convenience. We also included a clean, updated version of the manuscript, tables, figures, and supplementary material in the attached files.

Once again, we greatly appreciate the review feedback and comments.

Reviewer 1

Thank you for reviewing the section on infant neurobehavior. We found your comments to be very insightful.

We have updated the methods section to provide a more detailed explanation of the NNNS assessment. This highlights its role as a highly specific functional tool that evaluates the neurobehavioral repertoire of neonates across 12 distinct domains. This includes elaborating on the NNNS's clinical significance and ability to predict developmental outcomes, as demonstrated in numerous studies. In the discussion section, we have addressed the specific findings related to the NNNS scores in our study. Specifically, we emphasized the associations between phthalate metabolites and the downregulation of key neurotransmitter precursors, such as tyrosine and 5-hydroxytryptophan, and their subsequent negative impacts on attention and arousal scores. Furthermore, we highlighted the potential long-term developmental consequences of these associations, reinforcing the need for further research.

We hope these changes meet your expectations and clarify our research findings and their potential impact on infant neurodevelopment. Thank you once again for your constructive comments.

Line 200: reference 32 is incorrect for this statement and the NNNS was not developed by the NIH it was developed for the NIH.

Line 201, reference 33 is not the correct reference for the NNNS. It was published as a supplement to Pediatrics by Lester and Tronick.

Thank you for pointing out these errors. The correct citation was added, and the updated reference is now:

Lester, B. M. & Tronick, E. Z. History and description of the Neonatal Intensive Care Unit Network Neurobehavioral Scale. *Pediatrics* 113, 634–640 (2004).

Further, we have updated the language in lines 284-285:

Initially developed for the National Institutes of Health (NIH) to evaluate drug-exposed and at-risk infant, the NNNS is a validated measure of early neurodevelopment in very young infants.

Line 209, how does reference 35 relate to NNNS?

This reference in question is now reference 36 cited in line 292.

This was updated and clarified to refer to previously published research, which details the correlations between NNNS summary scores for this sample if the reader would like further information. The following sentence was added in lines 288-289 to clarify this point.

Further, research has indicated that NNNS scores are associated with differences in behavior in early childhood³⁴⁻³⁶.

PCA was used to derive attention and arousal scores. However, no information is presented about the PCA (percent variance accounted for, eigenvalues, item loadings etc)

Thank you for requesting expanded information on the PCA for the attention and arousal scores used as the outcome of this analysis. To address this comment, we have greatly expanded the NNNS score methods subsection to give readers more context on these measures. Specifically, we have added the following in lines 304-323:

The first composite score assesses newborn arousal by combining summary scores related to excitability, arousal, the number of calming strategies required by the examiner, physiological signs of stress (e.g., tremors, startles), and the newborn's self-regulation capacity (reverse scored). Higher scores on this composite indicate increased levels of newborn arousal. The second composite score measures the newborn's attention, reflecting their ability to respond to, focus on, and track external stimuli. This score is summarized by components related to attention and lethargy (reverse scored), with higher scores indicating more mature levels of newborn attention. In this study, standardized scores were created from the averages of each component. Previously published research has detailed the correlations between NNNS summary scores for this sample and demonstrated that the summary scores in each composite were moderately to highly correlated, supporting the creation of two overarching scores³⁷.

Figure 3 (a and b), how do we know what are the statistically significant effects and what do the colors mean? It looks as if the arousal associations are stronger than the attention findings in both a and b.

Thank you for this clarification on Figure 3 (a and b). This figure was designed to give readers a visualization of the significant, annotated findings from the meet-in-the-middle analysis. We expanded the figure title to include more information on what is displayed in this visual to avoid misinterpretation of the results.

Figure 3. Sankey plots (a) visit 1 (n = 81) and (b) visit 2 (n = 71). The first node is the phthalate exposures, leading to the associated metabolites in the second node, and ending at the NICU Net Neurobehavioral Scale (NNNS) outcomes, attention and arousal, in the last node. The yellow/blue colors on the left-hand side indicate different phthalate metabolites. The red colors on the right-hand side differentiate attention (dark red) and arousal (light red) or both (orange)

outcomes. Only significant, annotated findings are displayed in this figure; for details on the significance level and beta values, please see Table 4.

Lines 443-450: This section is a misunderstanding of the NNNS. It is, in fact, a “highly specific functional assessment” as it takes the entire neurobehavioral repertoire of the neonate and beaks it down into 12 functional domains. The NNNS has been related to chemicals, toxins and pollutants in a number of studies. And the NNNS does have clinical significance. It has been shown to predict developmental outcome in 17 studies.

It is hard to understand what are the exact NNNS findings in this study. This seems to be a missed opportunity because one of the strengths of this study could be that because the NNNS does predict developmental outcome these NNNS findings could suggest that phthalates could have long term developmental consequences for these infants and some of the potential metabolic pathways that are responsible.

Thank you for these important comments on the utility of the NNNS scores and how they could be better represented throughout this manuscript. We have added the following sentences to the NNNS methods subsection to provide more context on the clinical significance within Black populations specifically and more broadly.

In lines 286-287:

The measure has demonstrated validity across multiple studies, including those with US, Black participants^{33,34}.

In lines 292-293:

Further, research has indicated that NNNS scores are associated with differences in behavior in early childhood³⁴⁻³⁶.

We also added language to the discussion section, expanding on how these results might indicate long-term developmental consequences. We called for more research on the role of tryptophan and tyrosine metabolism to further understand what was observed in this study. Updated language can be found in lines 557-562.

The observed associations between phthalate metabolites and downregulation of key neurotransmitter precursors, coupled with negative impacts on attention and arousal scores, suggest that phthalates may have long-term developmental consequences for these infants. This study highlights the importance of further targeted research into tryptophan and tyrosine metabolism in both infants and children to understand their role in development.

We also updated the limitations section in lines 647-653 to alter the language to provide a more nuanced understanding to the reader of the utility of NNNS scores:

Using NNNS scores as an outcome presents certain considerations. The design of NNNS is intended to encompass a broad spectrum of neurobehavioral domains, including attention and arousal. While this lack of specificity might limit its precision in pinpointing specific clinical phenomena, it enables the test to capture general trends in infants. This broader scope facilitates comparability across infants within a study population⁷¹. Although NNNS scores may not provide clinical recognition of a specific condition, their ability to highlight general trends offers valuable insights into both present and future neurobehavioral function³⁴⁻³⁶.

Reviewer 2

Thank you for your detailed and insightful feedback on our manuscript. We have carefully considered and addressed your comments to enhance the clarity and rigor of our study. We acknowledged and corrected oversights in terminology and provided additional context for the study's design and limitations, including the exclusion criteria and how characteristics compare to the larger cohort. We also expanded on methodological details, such as the NNNS assessment conditions, the rationale behind using separate regression models, and the exploratory nature of the MITM analysis.

Detailed changes can be found below. We thank you again for your time in reviewing this work.

Introduction:

Line 82: The phrase "impact maternal serum samples" is not very clear. What about maternal serum samples was studied?

Thank you for asking for further clarification on the specifics of this study. We have updated the language to for clarity in lines 116-118.

One study examined the effects of phthalates on the maternal metabolome, analyzing serum samples obtained during the third trimester, as well as the impact of phthalates on the placental metabolome¹⁶.

Line 87: Rephrasing as "results indicated" or "authors found" would be more precise.

Thank you for this suggestion. The language was changed in lines 123 to "results indicated".

Line 91: The mechanisms under investigation don't strictly underpin exposure but its health effects.

We agreed that the language used previously in this sentence was confusing and have updated to reflect the mechanisms under investigation do not strictly underpin the exposure but the health effects of the exposure. Changes are reflected in lines 127-131.

Evaluating the biological mechanisms linking prenatal exposure to phthalates with alterations in the newborn metabolome is crucial for elucidating early-life metabolic perturbations associated with these compounds and the corresponding health effects. This is particularly important given the well-documented health impacts of phthalates, especially in underserved and vulnerable populations.

Lines 95-102: There are no objectives dedicated to understanding the association between phthalate exposure and NNNS scores. Has this been investigated/demonstrated in another study? If so, could this prior work be cited? If not, can the authors explain why this analysis is not being performed?

Thank you for pointing out this critical point of clarification in our logic. Various large-scale studies have established a link between phthalates exposure and impact to neurological impacts, and these studies are highlighted in the sections described below.

Background, lines 135-135: Based on these initial metabolomics results and previous epidemiological research¹³⁻¹⁵, we also hypothesized that urinary phthalate levels may interfere with infant neurodevelopment (evaluated by NNNS scores) through perturbations in the newborn metabolome.

These studies were emphasized in the background section of this manuscript in lines 100-112:

Previous research has shown a link between prenatal urinary phthalate levels and impacts on neurobehavioral development in low-risk infants^{13,14}. Specifically, prenatal exposure to phthalates and its interaction with the maternal endocrine system has been associated with Apgar scores 5 minutes after delivery¹⁵. Further, Yolton, et al. found that prenatal di(2-ethylhexyl) phthalate (DEHP) metabolite levels measured at 26 weeks gestation were associated with nonoptimal reflexes, measured using NICU Network Neurobehavioral Scale (NNNS) scores, in 5-week-old male infants in a cohort of 350 mother/infant pairs¹³. Similarly, Kim, et al. found that DEHP metabolite levels measured in the third trimester were inversely associated with mental and psychomotor indices (Bayley Scales of Infants Development 2nd edition) measured at 6 months old in a cohort of 460 expectant mothers¹⁴. Understanding how intermediate biomarkers might underpin these associations could inform causal links or serve as metabolic markers for early intervention.

Methods:

Line 110: What is MWAS? Metabolome Wide Association Study? Spell out on first use.

We appreciate pointing out this oversight in the definition of an acronym. We have added the definition to line 151.

The study participants included in the metabolome-wide association study (MWAS)...

Lines 107-12: Inclusion/exclusion criteria should be reported in more detail. Can you elaborate specifically on what chronic medical conditions and what medications were used as an exclusion during recruitment? It seems like there were a lot of restrictions at recruitment, which affects generalizability and selection bias, especially with the increased prevalence of chronic medical conditions in AA/Black communities.

Thank you for this comment. We added more language providing context for the decision to exclude those with chronic medical conditions and medications (lines 147-150) and added this caveat to the limitations paragraph of the discussion section (lines 555-557).

Lines 147-150: As the primary goal of this cohort was to identify what may be contributing to adverse birth outcomes in African Americans in otherwise healthy women, participants were excluded if there was any active diagnosis of chronic medical conditions or chronic use of prescription medications (verified through the prenatal medical record).

Lines 604-606: This study was limited to US born participants without active chronic medical conditions or chronic medication use, limiting the generalizability of these results to populations with these characteristics.

Do the authors have any information on ethnicity (i.e., Hispanic/Latina Black women)?

Thank you for this point of clarification. This cohort was people who self-identified as non-Hispanic African American or Black. We have added this to the methods section, lines 146.

Participants were eligible for the study if they self-reported non-Hispanic African American or Black race.

Lines 124-125: The authors stated “Previous research has demonstrated that a one-time urine sample can capture a representative picture of recent phthalate exposure, even with the relatively short half-life (around 12 hours)” – This may be an overstatement, I think there are still limitations due to potential variations in exposure levels throughout pregnancy. Levels can vary based on changes in dietary/supplemental intake and lifestyle changes due to pregnancy. It would be good to mention this as a limitation in the Discussion.

Thank you for this comment. We added to the limitations section that we lack information on diet, which would have been an important consideration as much of the phthalate exposure comes from food and drink packaging.

We also added that phthalates can still vary over time and a one-time sample might misrepresent an individual’s overall exposure experience.

Lines 606-609:

There were also limitations to consider with the covariate set. First, there was also no information on diet available, which would be an important covariate with phthalates, as much of the human exposure is from food and drink packaging¹. Future studies looking at phthalate exposure should collect this information.

Lines 615-618:

Beyond covariates, the phthalate measurement was a one-time snapshot of an exposure level that is dynamic and can change over time, which could misrepresent an individual’s overall exposure experience, although research indicates a one-time sample can be representative¹⁹.

Line 130: Check spelling of MECCP.

Thank you for catching this error. The spelling was corrected in line 195.

Lines 161-2: What is the stability of the metabolomic signatures at 2-8C and in the 0-3 month range that represents the storage conditions prior to long-term storage at -80C?

Thank you for this important consideration to the metabolomics profile. Although we do not have information on to comment specifically on the stability of the metabolomic signatures prior to long-term storage at -80C, we added a sentence to the methods section recognizing the potential for some signal degradation, lines 241-244.

Samples were stored in a walk-in refrigerator (2-8 °C) for up to three months until they were transferred to Emory University and stored within gas-impermeable bags with dessicant in a freezer (-80 °C) until further processing. The site used consistency in collection, storage, and shipment of biospecimens and any signals that quickly degrade, or that would have variability in

degradation due to sample storage time, would not be expected to arise as significantly associated with the study phenotype.

Line 164: Is there information available from the researchers or prior literature to understand how well data from a single 15-mm punch represent the measured features with respect to an entire blood spot?

We appreciate your thoughtfulness on ensuring transparency around the dried blood spot samples and have expanded the limitations paragraph in the discussion section to emphasize this point. Changes are highlighted below and can be found in lines 640-644.

Regarding newborn DBS samples obtained through heel-sticks, issues have been pointed out, such as uneven cell distribution, cell lysis, hematocrit variation, and pre-analytical clotting when employing untargeted metabolomics analysis on this biomatrix⁶⁸⁻⁷⁰. Despite the normalization of untargeted DBS metabolomics analysis based on the size and weight of the DBS, limitations in sample volume prevented the application of additional, potentially more optimal normalization methods, such as adjustments for hemoglobin, specific gravity, protein, or potassium⁷¹. Further, as this investigation used a single, 15-mm punch, this sample may not represent larger portions of blood because of potential variation in signals dependent on spotting and drying time.

Line 199: In this section, could the authors please indicate how many infants were assessed with the NNNS? If the number of infants is smaller at age of NNNS assessment than at exposure assessment, could the authors summarize the main reasons why NNNS could not be administered at the appropriate age?

Line 202: Could the authors indicate what other developmental measures the NNNS correlates with at later ages? In other words, how reliable is the NNNS in predicting developmental assessment scores in later infancy?

Lines 207-8: Please provide more information on the assessment and its circumstances. Where was NNNS administered and in what conditions? How many research specialists were involved in administering it? Were caregivers present and what was their role? Also, please provide more information on the NNNS domains, what they measure and how they are assessed. Perhaps the latter could be provided as online supplemental material. Finally, newborns change developmentally between 2 and 4 weeks of age. Could the authors state whether the entire assessment is administered regardless of age or are there age-specific start points?

Thank you for these points of clarification. Information answering each of these points has been added to the methods section in lines 291-319.

In this study, the NNNS was administered when infants were on average 2 to 4 weeks old (corrected for gestational age at birth) by trained and certified research specialists among those whose families agreed to participate in the infant follow-up. The NNNS administration did not vary across the 2- to 4-week age range. The NNNS was administered in a quiet location in the participant's home or the lab setting, according to standardized procedures. The measure was administered by one research specialist with a second research specialist present at the visit filming the administration for reliability checks. Both examiners were blind to the newborn's exposure group. All examiners completed a 5-day training program and established adequate inter-rater reliability with a NNNS trainer. Caregivers were present in the home or the lab setting but did not participate in the administration of the NNNS.

NNNS standardized arousal and attention summary scores were created on the basis of confirmatory principal components analyses, and were used in the statistical analyses. The first composite score assesses newborn arousal by combining summary scores related to excitability, arousal, the number of calming strategies required by the examiner, physiological signs of stress (e.g., tremors, startles), and the newborn's self-regulation capacity (reverse scored). Higher scores on this composite indicate increased levels of newborn arousal. The second composite score measures the newborn's attention, reflecting their ability to respond to, focus on, and track external stimuli. This score is summarized by components related to attention and lethargy (reverse scored), with higher scores indicating more mature levels of newborn attention. In this study, standardized scores were created from the averages of each component. Previously published research has detailed the correlations between NNNS summary scores for this sample and demonstrated that the summary scores in each composite were moderately to highly correlated, supporting the creation of two overarching scores³⁷.

Statistical analysis:

Please mention the number (%) of women lost to follow-up and excluded from the analysis due to missing data. In fact, it would be useful if the authors provided a flow diagram for the selection of participants into the study.

We appreciate this suggestion and have created a population selection flowchart that has been included in the supplementary materials (Figure S1).

It is unclear whether the authors assessed for potential selection bias. Were characteristics of excluded women similar to those included?

Table 1: how do the study samples compare on key sociodemographic characteristics (and exposure markers) to those who are excluded from analysis?

Thank you for these important considerations. To address these comments, we created Table S2. Results from this are highlighted in lines 404-405.

Characteristics from both subsets did not differ from the larger Atlanta African American Maternal Child Cohort (Table S2).

The authors mentioned using a DAG to identify confounders. I recommend adding this DAG as a figure.

Thank you for this recommendation. We agreed that it would be important to share this with the readers, and the DAG is now included in the Supplementary figures (Figure S2).

Covariates were retrieved from medical records, but it is not too clear what questions were asked, as well as the time frame. Especially questions about tobacco, marijuana, and alcohol use – Was this assessed during pregnancy? Or past use? Because this is a self-report and there is no information on frequency, intensity, and duration of use, there is definitely room for residual confounding and misclassification. Further, because of social desirability, women may have underreported their use – please mention these limitations in this discussion.

We appreciate your request for increased transparency around how covariates were collected. We have expanded the methods section to provide more details and have also included the suggest limitation in the limitations paragraph of the discussion section.

Methods section, lines 177-181:

During the first visit in early pregnancy, sociodemographic data such as maternal age and education were collected from self-reported surveys and from prenatal administrative records review. Health information was completed by self-report, including prior pregnancies (if any) and substance use. For substance use, medical records were reviewed for reported use prior to pregnancy, and use during pregnancy was self-reported.

Discussion section, lines 606-612:

Next, there was no information available from the substance use variables on frequency, intensity, and duration of use, information future studies could collect to prevent residual confounding. Further, as this is a pregnancy cohort, women may have underreported their use due to social desirability bias, introducing the potential for misclassification bias. However, reported use of alcohol and tobacco in this population are only slightly lower than national estimates and use of marijuana are higher than national estimates⁶⁶, so we hypothesize any effect from this bias would be minimal.

Line 212: Could the authors provide some justification for this choice when exposure metrics are available at two points? Given interest in critical windows, two separate regression models weaken any conclusions to that could be made to that effect. Furthermore, when the metabolomic profile is only measured at one time following both exposures, some thought needs to be given to the fact that the levels/profiles of the features may not related to the two exposures independently.

Thank you for your insightful feedback. We acknowledge that evaluating exposure metrics at two distinct points using separate regression models may not fully capture the independent effects of each exposure time point. However, our decision to use one timepoint for the metabolomic profile was driven by the complexity of the methods involved and the need to ensure robust and accurate data analysis. Further, unlike persistent organic pollutants, phthalates metabolites are short-lived and its levels vary across different pregnancy time windows. Thus, we chose to use the current analytic approach which allows us to examine the critical time window of exposure (i.e., early vs late pregnancy) and their independent impact on newborn metabolome in relation to infant neurodevelopment.

By focusing on a single timepoint for metabolomic profiling, we aimed to maintain methodological consistency and analytical clarity. Despite this, we believe that our approach still provides valuable insights into critical windows of exposure.

To address this point further, we have added a sentence to the methods section and a consideration in the limitations section, where we discussed the need for future studies to apply longitudinal study design and more sophisticated approaches to independently assess the effects of multiple exposure time points on the metabolomic profile.

Methods, lines 321-325:

We analyzed phthalate metabolite data from the two study prenatal visits cross-sectionally. The use of two separate regression models allows us to examine the potential differential impacts of phthalate exposure during early and late pregnancy, thereby offering preliminary indications of critical periods without the added complexity of a longitudinal analysis that might compromise the integrity of our analysis.

Discussion, lines 622-626:

Additionally, this research considered each timepoint cross-sectionally. Future studies should consider more sophisticated approaches to independently assess the effects of multiple exposure time points on the metabolomic profile. However, our current approach serves as a foundational step in understanding the temporal dynamics of phthalate exposure and its impact on the newborn metabolome.

Line 235: Could the authors give some indication of how many features were significant at the FDR-corrected $q < 0.2$ threshold and what implications the less stringent p-value cutoff had for the pathway analysis? How did loosening the cutoff affect number of features included in the study?

Thank you for this clarification question. The number of features at the different thresholding levels is summarized in Table S4. A callout to this table was added to line 348.

A common limitation of the pathway analysis tools used in metabolomics investigations is the requirement of at least 50 significant features to run the bioinformatics tool effectively. Given that this study is hypothesis-generating, we opted to use a less stringent p-value cutoff to increase the number of features available for analysis. This approach allowed us to explore a broader range of potential metabolic perturbations and generate more comprehensive hypotheses. Nonetheless, we acknowledge this limitation in the discussion section, where we address the potential implications of using a less stringent cutoff and emphasize the need for further validation in future studies with more robust statistical criteria:

Discussion (lines 629-633):

Unfortunately, for the MWAS pathway analysis and the entirety of the MITM analysis, we could not use FDR-corrected significance thresholds due to the limited statistical power resulting from the sample size, as the more stringent threshold would not include a sufficient number of metabolic features to conduct a meaningful pathway analysis or have MITM results. Therefore, these results should be carefully interpreted and confirmed using additional studies with larger sample sizes.

This analytic strategy is in line with numerous existing metabolomics research^{6,7}.

Were these metabolite groups analyzed separately to account for the level of similarity/certainty in identification?

Thank you for your comment and clarification. We did not analyze metabolites by groups initially because the pathway analysis tool only provides putative annotations based on m/z values, representing the lowest confidence level in metabolite identification with the moderate-to-high likelihood of false positive discovery in the chemical annotation. Therefore, we decided against grouping these metabolites for analysis at that stage. Instead, we performed group analysis only

after achieving higher confidence in chemical annotation and matching. This approach ensures more accurate and reliable results by focusing on metabolites with verified identities.

Line 269: Review text.

Thank you for pointing out this typo. The language of this sentence was altered for clarity in line 382.

First, the population used in the larger MWAS was subset to those with the NNNS score outcome available (Visit 1 n=81; Visit 2 n=71).

Line 277: This is the first time that the exploratory nature of the MITM is mentioned. Could the authors clarify this in the abstract as well?

Thank you for this helpful suggestion. We added the word “exploratory” to both the abstract and to the background sections of the paper to improve clarity.

Abstract:

Then, an exploratory meet-in-the-middle (MITM) analysis was conducted...

Background, line 137:

To test this hypothesis, we conducted an exploratory metabolomics analysis to further examine the potential mechanisms behind prenatal exposure to phthalates and neurobehavioral outcomes by identifying potential intermediate biomarkers between phthalate exposure and NNNS scores within a subset of the same population using a meet-in-the-middle (MITM) approach.

Results

Lines 293-4: Could the authors provide levels of correlation among phthalate metabolites at each time and across time?

Thank you for requesting this additional information. Correlation plots were created and can be found in Figure S3. Additionally, we have added language in the results section of the manuscript in lines 421-423.

Correlations of phthalate metabolites within and across visit are summarized for the full sample (Figure S3). Within visits, only phthalate metabolites that are related to DEHP (MEHP, MEOHP, MEHHP, and MECPP) were highly correlated. Across visits, phthalate metabolites were not highly correlated.

Line 310: Only attention and arousal are mentioned as outcomes yet the NNNS was stated to yield 12 sub-domains (lines 203-4). It is not clear from the methods that attention and arousal will be the main endpoints under study. Could the authors clarify this in the methods section and provide some justification for this choice?

We have greatly expanded the NNNS subsection in the methods section of the manuscript and have added clarity around how the attention and arousal scores were created in lines 304-323.

The first composite score assesses newborn arousal by combining summary scores related to excitability, arousal, the number of calming strategies required by the examiner, physiological signs of stress (e.g., tremors, startles), and the newborn's self-regulation capacity (reverse scored). Higher scores on this composite indicate increased levels of newborn arousal. The second composite score measures the newborn's attention, reflecting their ability to respond to, focus on, and track external stimuli. This score is summarized by components related to attention and lethargy (reverse scored), with higher scores indicating more mature levels of newborn attention. In this study, standardized scores were created from the averages of each component. Previously published research has detailed the correlations between NNNS summary scores for this sample³⁷.

Discussion:

Line 353: There needs to be some caution regarding this conclusion b/c the analyses from visit 1 and visit 2 do not include the same set of participants, and furthermore, the number of participants was much reduced from MWAS to the MITM analysis.

Thank you for this insightful comment. We have expanded the limitations paragraph in the discussion section to help readers interpret the results with appropriate caution. Additionally, we enhanced the methods section to provide greater transparency regarding the composition of the cohort, including detailed information on the overlapping and independent numbers of participants across different visits.

Methods 150-154: The study participants included in the metabolome-wide association study (MWAS) were further restricted to mother-newborn pairs with mothers who had urine samples available for the determination of urinary phthalate levels and newborns with metabolomics profiles from dried blood spot samples (visit 1 n = 216; visit 2 n = 145; Figure S1). Participants could be in visit 1 only (n=76), visit 2 only (n=5), or represented in both visits (n=140).

Discussion lines 631-639: This was addressed by imposing stringent criteria whenever possible throughout the analysis, including applying the Benjamini-Hochberg FDR. Unfortunately, for the MWAS pathway analysis and the entirety of the MITM analysis, we could not use FDR-corrected significance thresholds due to the limited statistical power resulting from the sample size, as the more stringent threshold would not include a sufficient number of metabolic features to conduct a meaningful pathway analysis or have MITM results. Therefore, these results should be carefully interpreted and confirmed using additional studies with larger sample sizes.

Lines 388-90: This is an interesting point but brings up the question of what the range of tyrosine/tryptophan values observed in this study represents. Can they be viewed as being within a normative range? Furthermore, at what ages in references 49 & 50 were the clinical correlates of tyrosine metabolic pathways observed? Clinical disease is likely associated with values outside of a typical range and it is unclear how strongly the attention scores on NNNS correlate with developmental measures in later childhood.

Thank you for highlighting this important point. Due to budget constraints, we were unable to conduct targeted quantification of the identified metabolites, as this study was designed as an untargeted metabolomics analysis for hypothesis generation. Consequently, we believe linking these preliminary findings directly to clinical outcomes would be an overstatement. To avoid misinterpretation and address this comment, we have removed the sentence in question for clarity.

Another strength of this study is the temporality between exposure and outcome due to the prospective cohort design.

We greatly appreciate pointing out this strength and agree that it should be added. This is now in the strengths paragraph of the discussion section, lines 595-596.

As this is a prospective cohort design, we also had well established temporality between the exposure and outcome.

One of the criteria for eligibility was US born Black women, which limits generalizability to non-US born Black women – it would be important to discuss this as a limitation.

Thank you for this comment - we have added a sentence with considerations around the generalizability of these results, lines 605-607.

This study was limited to US born participants without active chronic medical conditions or chronic medication use, limiting the generalizability of these results to populations with these characteristics.

Reviewer 3

Main revisions:

Line 175: The authors should better explain the population enrolled in the study. It is not clear to the reader how many women are involved in the first and/or in the second visit. In the abstract, it is reported that 278 mother-newborn pairs have been under investigation, however only 216 participants were reported for visit 1 and 145 for visit 2. I suppose all 145 participants to the 2 visit were attending the first visit. Or not?

Thank you for pointing out the discrepancy in the abstract. This was an error and has been corrected:

We quantified eight phthalate metabolites in prenatal urine samples collected between 8- and 14-weeks' (visit 1; n=216) and 24- and 30-weeks' gestation (visit 2; n=145)

We also created a population flow chart to increase transparency on how the population was selected from the larger cohort. We also provided details on how many participants overlap, are in just visit 1 or just visit 2. These changes can be found in the methods section, lines 150-154.

The study participants included in the metabolome-wide association study (MWAS) were further restricted to mother-newborn pairs with mothers who had urine samples available for the determination of urinary phthalate levels and newborns with metabolomics profiles from dried blood spot samples (visit 1 n = 216; visit 2 n = 145; Figure S1). Participants could be in visit 1 only (n=76), visit 2 only (n=5), or represented in both visits (n=140).

Line 324-325 and Table S3: Exposure to phthalates is very similar between the first and second visits, as shown in table 2. How do you explain such a higher number of altered metabolites related to visit 1? In the discussion, the Authors proposed the importance of timing in exposure, however, the hypothesis should be better explained, by proposing possible mechanisms that protect or affect the newborns' metabolism. Specifically, this aspect should be explored for phthalates such as MEHP, MECPP, and MEOHP which

appear to be correlated only in one of the two visits, despite having been measured in both. Likewise, it should be better evaluated why there is such a variable number of metabolic features associated with MBP (as reported in Table S3), especially given its relevance in the context.

Thank you for this comment. To address the feedback on timing, we have added the following paragraph to the discussion section of the manuscript, lines 493-502.

These findings, along with the observed variances in metabolite perturbations across visits 1 and 2 and the differential associations of various phthalate metabolites with distinct gestational stages, suggest potential implications for fetal development. In early pregnancy (visit 1), major development of the central nervous system, eyes, teeth, genitalia, and refinement of the limbs and heart are taking place. Whereas later pregnancy (visit 2) primarily entails refinement processes within these systems, particularly focusing on central nervous system maturation and refinement within the eyes and genitalia¹. The absence of significant impacts on later gestational periods does not preclude potential effects during the critical phases of early embryogenesis. These findings underscore the necessity for targeted investigations spanning the entirety of gestation and advocate for nuanced consideration of pregnancy timing when evaluating environmental exposures.

To examine correlations within the exposure, correlation plots were created and can be found in Figure S3. Additionally, we have added language in the results section of the manuscript in lines 421-423.

Correlations of phthalate metabolites within and across visit are summarized for the full sample (Figure S3). Within visits, only phthalate metabolites that are related to DEHP (MEHP, MEOHP, MEHHP, and MECPP) were highly correlated. Across visits, phthalate metabolites were not highly correlated.

To address the comment on MBP, we noted that many of the highlighted metabolites were associated with the MPB phthalate metabolite. These edits can be found in the discussion section, lines 509-515.

Both were associated with an upregulation in response to MBP exposure, potentially as a protective response. Additionally, the presence of thyroxine, a thyroid hormone, is noteworthy, as low levels of this hormone have been associated with increased susceptibility to illness and neurodevelopmental issues in newborns⁴⁸. This was downregulated in response to MBP exposure. Finally, the metabolite dihomo-gamma-linoleate within linoleic acid metabolism was associated with a downregulation by MBP exposure.

Table 3, Table 4, and discussion: Various metabolites belonging to the exposome have been correlated with phthalates. If these molecules are, for example, taken from the diet, the authors should propose a possible role of phthalates in the altered metabolism, absorption, or excretion of these molecules. Or not?

Thank you for this important interpretation of the results from the untargeted metabolomics study. After consideration, we agreed that this would be something important to highlight in the discussion section and wrote a new paragraph, lines 538-541.

We also identified metabolites in the MWAS, including 3-hydroxyhippuric acid and hydroxyphenylacetic acid, which are related to the microbiome^{64,65}. Metabolomics has the

potential to highlight areas for further exploration within the microbiome, offering insights into how microbial composition and activity may influence metabolic pathways.

Minor revisions

Line 73 DEHP: write the full name.

Thank you for pointing out this oversight in acronym definition. We have updated the text in line 105.

Line 112 and 197: The Authors referred to serum metabolomics, which probably meant blood metabolomics. Or not? Moreover, the authors stated “participants who had urine and blood samples”: Do they mean mother-newborn pairs?

We appreciate pointing out the inaccurate language used in this sentence, and we have revised for clarity. Updates can be found in the methods section, lines 150-154.

The study participants included in the metabolome-wide association study (MWAS) were further restricted to mother-newborn pairs with mothers who had urine samples available for the determination of urinary phthalate levels and newborns with metabolomics profiles from dried blood spot samples (visit 1 n = 216; visit 2 n = 145; Figure S1).

Line 116: The Authors evaluated if women under study followed some specific diet? As reported in the literature, phthalate exposure is linked to food and beverage consumption, but it has not been properly underlined in the background neither in the study population.

Thank you for pointing out this limitation to the design of the study. We have expanded the limitations paragraph in the discussion section to discuss this point, lines 587-594.

There were also limitations to consider with the covariate set. First, there was also no information on diet available, which would be an important covariate with phthalates, as much of the human exposure is from food and drink packaging¹. Future studies looking at phthalate exposure should collect this information.

Line 119-122: The Authors considered the type of delivery? Some reported metabolites, as 3-Hydroxyhippuric acid or Hydroxyphenylacetic acid are related to the microbiome, which can be impacted by the type of birth.

Thank you for this important comment. We felt it was important to highlight and included this as a main point in the discussion section on the discussion around how findings in the metabolomics layer might point to other important areas to explore in other omics layers. The additions can be found in the discussion section, lines 541-545.

Although investigating the microbiome is beyond the scope of this research, it is crucial to consider factors such as the type of birth (vaginal or cesarean) when studying newborn populations. These factors can significantly impact the microbiome and subsequent metabolic profiles. Therefore, future studies should ensure sample sizes are sufficiently large to allow for stratification based on these variables.

Line 150: In the manuscript, the LOD of each phthalate has been reported. It would be of interest to also report the LLOQ, given that the phthalates have been quantified, the linearity ranges of the curve, and possibly the accuracy range considered and %CV.

Thank you for asking for these additional details on the phthalate metabolites. We have provided these in an expanded phthalates paragraph in the methods section, lines 213-227.

As previously described, the limits of quantification (LOQ; ng/mL) for these phthalate metabolites were: 1.0 (MEP), 4.0 (MBP), 2.0 (MiBP), 0.2 (MBzP), 0.2 (MEHP), 0.4 (MEOHP), 0.4 (MEHHP), and 5.25 (MECPP)¹⁷. The linear range was 0.01-50 ng/mL for all phthalate metabolites expect for MEP, which was 5-500 ng/mL. Samples that fell above those concentrations were diluted and repeated. Accuracy based upon National Institute of Standards and Technology (NIST) were 91-105%. The percent relative standard deviations were all below 12%.

Line 316-317: The sentence “There were 28 perturbed metabolic pathways in visit 1 only and 12 in visit 2 only” could be changed to: “There were 28 perturbed metabolic pathways associated with phthalate measured in visit 1 and 12 in visit 2.”? As it is written, it may be confusing for the reader. The Author are speaking of perturbed metabolic pathways associated with phthalates at this 2 specific time points.

Thank you for this suggested language change. We have updated the sentence as suggested, lines 434-435.

There were 28 perturbed metabolic pathways associated with phthalate measured in visit 1 and 12 in visit 2.

Line 370-371: Probably due to a typo, dihomo-gamma-linoleate was defined as an essential amino acid instead of polyunsaturated fatty acid.

Thank you for identifying this error. This has been corrected in the discussion section, line 509.

This polyunsaturated fatty acid is a component of neural membranes⁵⁰.

Figure 3: visit 2 (n=1)! Is n=1 correct? Maybe n=71?

Thank you for pointing out this typo in the legend. We have corrected from n=1 to n=71.

Table 2: It was not reported what the symbol “#” stands for.

To address this oversight, we added this information to the footer of Table 2:

*Secondary metabolite; GM = geometric mean; # = Not calculated: proportion of results below limit of detection was too high to provide a valid result.

Table 3: The 3-Methylhippuric acid is a urinary metabolite, as correctly reported. Is it usual to find it in the new-born blood? It should be a xylene metabolite.

Thank you for pointing out this incorrect interpretation of the findings in Table 3. The suggested change has been made.

Table 4: Cotinine is a biomarker for tobacco use. 13-14% of mothers smoke (calculated from Table 1). Only metabolites present in at least 95% of the DBS samples were taken into consideration. Then, how is it possible that Cotinine is present in the table? The mass 176.095 has been related to MECPP, MEHP and Σ DEHP in visit 1, while to MiBP in visit 2. Could you explain this?

In untargeted metabolomics studies, the accurate identification of metabolites can be challenging due to the potential overlap of mass-to-charge ratios (m/z) among different compounds. Cotinine, a biomarker for tobacco use, and serotonin, a neurotransmitter, share a similar mass (176.095), making it difficult to distinguish between them using liquid chromatography-mass spectrometry (LCMS) alone.

Additionally, cotinine may be present in our samples not only as a result of direct smoking by the mothers but also due to environmental exposure to tobacco smoke. This possibility aligns with the observation that cotinine is present in a significant proportion of the samples, despite the relatively low percentage of mothers reporting smoking.

To clarify this point, we have added the following language to the discussion section to provide the reader with a more nuanced understanding underscoring this point, lines 564-570.

It is important to consider the interpretation of these findings within the context of exposure to complex mixtures. When conducting our statistical modeling, we considered each phthalate independently as surrogates of exposure. Thus, an observed metabolic perturbation associated with a particular phthalate may not necessarily indicate a causal association between the metabolic feature and that specific modeled indicator. Instead, such change may be more likely associated with multiple, correlated pollutants within a complex exposure mixture. This consistency in perturbed metabolites across different studies with varied single exposures suggests that future research should consider the cumulative impact of multiple exposures, integrating a mixtures analysis approach⁶⁰⁻⁶².

Table S3 and discussion: The MBP had a relevant role in metabolic perturbation (evident in Table S3) and it has been correlated in both visits. The authors should better clarify its role and possible differences from its analogues.

Thank you for pointing out this point to highlight. We have added the following sentence to the discussion section to address this comment. The addition can be found in the discussion section, lines 510-514.

The interplay of metabolites within these pathways emphasizes the potential impact of phthalates, specifically MBP, on abnormal neurodevelopment, underlining the need for closer examination of this exposure and its intricate biochemical influences on brain maturation during this critical developmental period.